# Foliar Co-Applications of Nitrogen and Iron on Vines at Different Developmental Stages Impacts Wine Grape (*Vitis vinifera* L.) Composition

**DOI:** 10.3390/plants13162203

**Published:** 2024-08-09

**Authors:** Xiaoke Fu, Xi Chen, Yiwen Chen, Yueran Hui, Rui Wang, Yaqi Wang

**Affiliations:** School of Agriculture, Ningxia University, Yinchuan 750021, China; xiaokefu@foxmail.com (X.F.); cxi2023@126.com (X.C.); 12022131382@stu.nxu.edu.cn (Y.C.); 15769573316@163.com (Y.H.); amwangrui@126.com (R.W.)

**Keywords:** nitrogen, iron, amino acid, flavonoid, wine grape quality

## Abstract

The co-application of N and Fe can improve wine grape composition and promote the formation of flavor compounds. To understand the effects of foliar co-application of N and Fe on wine grape quality and flavonoid content, urea and EDTA-FE were sprayed at three different developmental stages. Urea and EDTA-Fe were sprayed during the early stage of the expansion period, at the end of the early stage of the expansion period to the late stage of the veraison period, and during the late stage of the veraison period. The results demonstrated that the co-application of urea and EDTA-Fe, particularly N application during the late stage of the veraison period and Fe application during the early stage of the berry expansion period (N3Fe1), significantly improved grape quality. Specifically, the soluble solid content of berries increased by 2.78–19.13%, titratable acidity decreased by 6.67–18.84%, the sugar-acid ratio became more balanced, and yield increased by 13.08–40.71%. Further, there was a significant increase in the relative content of amino acids and flavonoids. In conclusion, the application of Fe and N fertilizers at the pre-expansion and late veraison stages of grapes can significantly improve the quality and yield of berries; ultimately, this establishes a foundation for future improvement in the nutritional value of grapes and wine.

## 1. Introduction

*Grapes* (*Vitis vinifera* L.) are one of the most important fruit crops grown worldwide due to their yield and economic value. In China, *grapes* hold significant cultural and economic importance, contributing extensively to both domestic consumption and export markets. The quality of *grape* berries is closely related to the balance between primary and secondary metabolites, which are essential for producing high-quality wines. Primary metabolites, such as glucose and fructose, are the main sugar compounds in *grapes*. High sugar accumulation in berries enhances the volatility of aromatic compounds, which is crucial for the wine’s flavor profile [1]. The amounts of these metabolites can be influenced by factors such as variety, harvest time, and berry sanitation [2,3]. Secondary metabolites, including tannins, anthocyanins, total phenols, and flavonoids, are primarily distributed in the pericarp and seed coat of *grapes* [4]. These compounds have antioxidant functions, protecting *grapes* from ultraviolet radiation and pathogens, and play a significant role in determining the quality of red wine; additionally, these metabolites also play an important role in the quality parameters of red wine [5,6].

Fertilization practices profoundly impact *grape* quality. The application of fertilizers affects various quality parameters, including berry size, sugar content, acidity, and the concentration of phenolic compounds. Mineral nutrition significantly affects *grape* yield and quality [7]. Among all nutrients, nitrogen (N) is essential as it forms part of major biological molecules, including chlorophyll, amino acids, nucleic acids, and hormones. These molecules significantly impact the nutritional growth, yield, metabolism, and energy production of *grapes* [8]. Insufficient N availability due to improper fertilization can hinder shoot growth, reproductive shoot development, and seed setting rate, ultimately affecting the growth, development, and yield of grapevines [9]. Conversely, proper N fertilization can significantly increase leaf N content, dry matter quality, and yield [10]. 

Iron (Fe) is another crucial micronutrient for grapevines. It acts as a cofactor or component of many enzymes involved in electron transfer and redox reactions [11]. Fe is vital for various physiological processes, such as photosynthesis, respiration, enzyme activation, chlorophyll biosynthesis, carbon and nitrogen assimilation, and phospholipid synthesis [12]. Therefore, Fe fertilization can influence fruit quality factors and yield in many fruit trees [13]. Numerous studies have demonstrated the benefits of foliar Fe spray on vineyard yield and berry sugar content [14]. However, traditional *grape* cultivation often relies on macronutrient fertilizers, neglecting the importance of micronutrients. Moreover, there is a lack of studies evaluating the variety, application rate, and timing of Fe fertilizers.

The synergetic application of macronutrient and micronutrient fertilizers is an important direction in plant nutrition research. This approach can promote the uptake of nutrient-rich elements in crops, improving crop yield and quality. Specifically, the coordinated application of N and Fe fertilizers, where some N forms interact with Fe uptake, can enhance crop yield and quality and alleviate Fe deficiency symptoms in plants [15,16]. However, the effects of N and Fe fertilization on soluble sugars, total phenols, and the antioxidant capacity of *grape* berries are not well understood. Additionally, there is a paucity of studies examining the impact of combined N and Fe fertilization at different growth stages on *grape* berry composition.

This study evaluates the combined effects of N and Fe fertilization on *grape* berry composition across different growth stages. Unlike previous research that primarily focuses on either macronutrient or micronutrient application, this study integrates the synergetic application of both N and Fe. By doing so, it provides a holistic understanding of how these nutrients interact and influence *grape* quality parameters such as soluble sugars, total phenols, and antioxidant capacities. Additionally, the study spans multiple developmental stages of grapevines, offering insights into the optimal timing for nutrient application to maximize berry quality and yield. This multifaceted approach not only addresses gaps in current literature but also has practical implications for improving vineyard management practices and enhancing the economic value of wine *grapes*. The primary aim of this research is to compare the effects of N and Fe co-application at different growth stages on *grape* physiological growth, berry composition, and flavonoid compounds, providing a foundation for the enhancement of wine *grape* quality through a comprehensive evaluation of N and Fe interactions across different growth periods.

## 2. Results

### 2.1. Effect of Co-Application of N and Fe on Photosynthetic Parameters of Wine Grape Leaves at Different Developmental Stages

As shown in Table 1, there were significant differences in the photosynthetic parameters of *grape* leaves from different treatments. The Pn content of *grape* leaves from treatments N1Fe1 and N3Fe1 was significantly higher than that from other treatments, with 13.35–14.2 μmol m^−2^ s^−1^ and 122.0–124.5 μmol m^−2^ s^−1^, respectively. The Gs content was higher in leaves from N1Fe1, N2Fe1, and N3Fe2 than in other treatments. The Tr of leaves from treatment N1Fe1 was the highest, reaching 3.73 vμmol m^−2^ s^−1^, while the Ci and WUE were the highest in leaves from the N3Fe1 treatment, at 332.0 μmol mol^−1^ and 6.48%, respectively. Overall, except for the interaction between N and Fe in leaf Tr, the application of N or Fe had a significant impact on *grape* leaf photosynthesis.

Further analysis of *grape* leaf chlorophyll content between treatment groups (Figure 1) revealed significant differences in chlorophyll a, chlorophyll b, and total chlorophyll. Chlorophyll a content was relatively higher in leaves from Fe1 treatment and reached a maximum value of 1.14 mg g^−1^ in leaves from treatment N2Fe1. Alternatively, chlorophyll b content was highest at 0.59 mg g^−1^ in leaves treated with N1Fe2, 11.32–68.75% higher compared to the remaining treatments. Overall, compared to the other treatments, the total chlorophyll content of the leaves of the N2Fe1 treatment was 1.66 mg g^−1^, which was higher than that of the N1Fe3, N2Fe2, N3Fe2, and N3Fe3, respectively, by 8.17–52.29%, 24.64–28.21%, 23.63–57.24% and 48.29–65.56%, respectively. In addition, the N3Fe2 and N3Fe3 treatments significantly increased the ratio of chlorophyll a to chlorophyll b compared to the other treatments.

### 2.2. Effect of Co-Application of N and Fe on N and Fe Contents in Leaves and Petioles of Wine Grape at Different Developmental Stages

The N and Fe contents in the leaves of wine *grapes* were significantly higher than in petioles (Figure 2). The total N content in *grape* leaves was significantly affected by the interaction between N and Fe fertilizer; the highest content of N was 32.49 g kg^−1^ in the N1Fe1 treatment, which possessed an average increase of 8.14% compared to the other treatments. Except for the highest total N content in petioles from the N3Fe1 treatment, there were no significant differences observed among the other treatments. Nonetheless, the total Fe content in the leaves and petioles of each treatment was significantly different; the total Fe content in the leaves of each treatment was as follows: N1Fe2 > N1Fe3 > N2Fe2 > N2Fe1 > N2Fe3 > N3Fe1 > N1Fe1 > N3Fe2 > N3Fe3; further, the total Fe content in petioles of each treatment was as follows: N2Fe2 > N1Fe2 > N3Fe3 > N3Fe2 > N1Fe3 = N2Fe1 = N2Fe3 > N1Fe1 > N3Fe1. The ratio of N content in leaves to N content in petioles was significantly higher under the N1Fe1 and N1Fe2 treatments than under the N2Fe2 and N3Fe1 treatments but was not expected to be significantly different from the treatments; however, the ratios of Fe content in leaves to Fe content in petioles varied significantly among the treatments. Overall, the N1Fe2, N1Fe3, and N3Fe1 treatments significantly increased the ratio of Fe content in leaves to Fe content in petioles compared to the other treatments.

### 2.3. Effect of Co-Application of N and Fe on N and Fe Contents in Leaves and Petioles of Wine Grape at Different Developmental Stages

As shown in Table 2, the effects of N and Fe co-application on the morphological indices of wine *grapes* were significantly different. The different application periods of N and Fe fertilizers significantly affected berry size, raceme length, and weight of the berries. Applying N and Fe fertilizers in the late stage of the berry veraison period significantly increased berry size, raceme length, and weight of berries. The berry size, raceme length, and berry weight from the N3Fe1 and N3Fe2 treatments were 6.32–40.14%, 19.81–48.77%, 1.52–8.29% higher than the other treatments, respectively. During the later stage of the veraison period, N application significantly increased berry yield and reached the peak value in the N3Fe1 treatment, which was 1.56 kg (yield plant) and 7355 kg ha^−2^ (yield), which increased by 13.08–40.71% compared to other treatments, followed by N3Fe2, N3Fe3, and N2Fe3.

### 2.4. Effect of Co-Application of N and Fe on Wine Grape Quality at Different Developmental Stages

The SSC of wine *grapes* from the N3Fe1 treatment was 2.78 to 19.13% higher than the other treatments (Table 3). The titratable acidity of wine *grapes* from the N3Fe1 treatment was 6.67 to 18.84% lower than the other treatments. Consequently, the SSC/TAC ratio was highest in all N3Fe1 treatments (50.42), representing an increase of 16.29–40.05% compared to the other treatments. The highest tannin content was observed in the N2Fe3 treatment, with an increase of 1.58–20.24% over other treatments. The N3Fe3 treatment yielded the highest anthocyanin content, with an increase of 16.19–45.05% relative to other treatments. Finally, the highest total phenolic content was found in the N2Fe3 treatment group, with an increase of 1.48–60.64% compared to the alternative treatments. 

### 2.5. Effect of Co-Application of N and Fe on Wine Grape Relative Content of Essential Amino Acids at Different Developmental Stages

The synergistic application of N and Fe at different stages had a highly significant effect on the relative content of amino acids in berries (Table 4), in which the N3Fe1 treatment showed a significantly higher content the content of L-Serine, L-Proline, L-Threonine, L-Aspartic Acid, L-Lysine, L-Histidine, L(+)-Arginine, Glycine, and L-Glutamic acid, and N3Fe2 treatment significantly increased the content of L-methionine, L-phenylalanine, L(+)-Arginine and L-Glutamic acid. In general, N application, Fe application, and the interaction of N and Fe had a significant impact on the contents of L-threonine, L-lysine, L-methionine, L(+)-Arginine, Glycine, and L-Glutamic acid. The changes in N application periods significantly or highly significantly affected the content of various amino acids in the berries, with the highest relative content at N3, followed by N2 and then N1; this indicated that supplemental nitrogen fertilization at the late stage of veraison significantly increased the amino acid content of *grape* berries.

### 2.6. Effect of Co-Application of N and Fe on Wine Grape Yield Flavonoids Content at Different Developmental Stages

Following differential flavonoid metabolite analysis of wine *grape* peel, 33 different flavonoid compounds out of a total of 46 were screened. The differences in the relative flavonoid content in *grape* peel at different periods of N and Fe co-application are shown in Table 5. There was no significant difference between the concentrations of silibinin, puerarin, rutin, cianidanol, dihydroxybenzoic acid, naringenin, myricetin, morin, daidzin, vitexin, icariin, troxerutin, L-epicatechin, isorhamnetin, genistin and procyanidin B2 in the peel under the conditions of N and Fe co-application at different stages. Nonetheless, N3Fe1 treatment significantly increased the contents of genistein, apigenin, baicalin, hesperetin, hesperidin, protocatechualdehyde, luteolin, diosmin, neohesperin, and artemisinin. N1Fe1 treatment significantly increased the myricitrin, hyperoside, and astragalin contents. Alternatively, N3Fe2 treatment significantly increased the content of quercetin and taxifolin rhamnoside.

## 3. Discussion

### 3.1. Physiological Growth of Wine Grape

Plants accumulate dry matter predominantly through photosynthesis; this accumulation can, then, directly affect their growth, yield, and quality [17]. Different soil moisture levels and N and Fe supplies can also affect the photosynthetic capacity, growth, and dry matter accumulation of plants, thus directly affecting their productivity [18]. Many studies have demonstrated the relationship between N fertilizer, Fe fertilizer, and photosynthesis [19,20], and some have established that appropriately delaying the nitrogen application period is conducive to improving the net photosynthetic rate of leaves, delaying leaf senescence, and prolonging photosynthetic time [21]. Additionally, the application of Fe fertilizer affects the stomatal structure of fruit tree leaves, therefore affecting leaf photosynthesis [22]. Overall, the present study demonstrated that co-applications of N and Fe application at different growth stages affected the photosynthetic index of leaves in wine *grapes*. The most suitable application period of N and Fe fertilizer was the application of N at the later stage of veraison and Fe at the early stage of expansion, aligning with prior studies [23]. 

Chlorophyll is an important component in plant photosynthesis, and its content can reflect the intensity of photosynthesis in functional plant leaves. Fe deficiency leads to changes in the chloroplast lamellae structure and a decrease in the number of chloroplast bases. In severe cases of Fe deficiency, it also leads to the disintegration of chloroplasts, thereby inhibiting photosynthesis [24]. It was established that, within a certain range, the chlorophyll content and photosynthetic rate of plant leaves are positively correlated with the N content of these leaves; in contrast, N supply imbalance could lead to the decline of photosynthetic capacity [25]. In the present study, the total chlorophyll content was highest when plants were under the treatment of N application from the expansion period to the veraison period and Fe application at the early stage of the expansion period. The results of this Fe treatment strategy corresponded with the understanding that Fe acts as a cofactor or component of various proteins and enzymes that are involved in the electron transfer system and reduction/oxidation reactions [11] and dominates some important physiological processes, such as photosynthesis, respiration, enzyme activation in the early stage of crop reproduction, and growth. In addition, early application of N and Fe may contribute to the initial establishment of photosynthetic mechanisms, whereas late application (N3Fe3) may enhance specific pathways associated with chlorophyll a synthesis more than chlorophyll b. The results of this study suggest that the use of N and Fe at early stages of growth may be more effective than that of chlorophyll b. This may be due to the fact that different growth stages have different nutritional requirements. At later stages of development, plants may prioritize chlorophyll a synthesis, which is more directly involved in the light reactions of photosynthesis. Higher chlorophyll A/b ratios may increase the plant’s ability to utilize available light more efficiently, which is critical during periods of high photosynthetic activity.

This study also found that the content of N and Fe in the leaves was the highest following N and Fe application in the early stage of the expansion period, indicating that the application of key nutrients on the leaf surface at the appropriate time of the growing season can directly or indirectly affect the internal solubility of nutrients. These findings align with a previous study that found that *grape* plants treated with 1% Fe-EDDHA combined with 1% urea possessed a higher N concentration than that in plants treated with only urea [26]. In addition, the application of N and Fe (N1Fe1) in the early stage of the expansion period significantly increased the ratio of N and Fe content in leaves and petioles, which may be related to the nutrient requirements of the developmental stages of the vine. During the early expansion period, leaf development has a higher demand for nutrients such as N, whereas, during the veraison period, the focus may shift to fruit development, affecting the distribution of nutrients. Therefore, early fertilization, especially with N, may enhance nutrient accumulation in the leaves, while strategic application of Fe may significantly increase Fe levels in the leaves, resulting in improved crop health and yield. 

### 3.2. Quality and Morphology of Wine Grape

Sugars and organic acids are the raw materials required for the synthesis of many other compounds; therefore, the content of these materials strongly affects the corresponding berry quality, thereby determining the taste and flavor of the fruit [27]. Within a certain range, leaves with high N content have been determined to be non-conducive to sugar accumulation in *grape* berries. Therefore, the proper application of N can improve the sugar content of berries [28]. Alternatively, Fe deficiency can lead to an increase in acid content and a decrease in sugar content in berries [29]. Results from the present study demonstrated that the soluble solid content of *grape* berries was the highest, and the titratable acidity content was the lowest with treatment of N at the late stage of the veraison period and Fe at the early stage of the expansion period. Overall, this indicated that the absorption and utilization of N and Fe by *grapes* at different growth stages were different; therefore, the application of N and Fe during the appropriate growth periods can promote an increase in the sugar-to-acid ratio in *grape* berries.

The tannin and total phenol contents were the highest when *grapes* were grown under the conditions of N application from the expansion period to the veraison period and Fe application at the later stage of the veraison period. Additionally, the anthocyanin content was the highest under the conditions of N and Fe application both at the later stage of the veraison period. Overall, these results can be attributed to the understanding that tannin and total phenols are affected by the interaction between N and Fe application periods, whereas anthocyanins are primarily affected by the Fe application period. *Grapes* require a large amount of anthocyanin during the veraison period, and the sugar formed by photosynthesis is an important substance for the synthesis of anthocyanin; therefore, Fe application can improve the efficiency of photosynthesis and promote the synthesis of anthocyanin [30,31]. In the present study, the application of N fertilizer at the later stage of the veraison period significantly increased the berry weight and yield of *grape* berries; this may be due to the increased nutrient demand of *grape* berries when they enter the veraison period. Therefore, applying nitrogen fertilizer at the later stage of the veraison period can provide sufficient nutrients for *grapes*, improve the quality of berries, and promote fruit ripening.

### 3.3. Amino Acids and Flavonoids of Wine Grape

Applying N fertilizer at the later stage of *grape* growth can increase the amino acid content in *grape* berries to meet the demand for N, promote the synthesis of secondary metabolites and aromatic substances, and effectively increase the content of important precursor amino acids of secondary metabolism [32]. Amino acids in wine *grapes* have been shown to be related to the formation of higher alcohols and esters, and N fertilizer can increase the content of 18 amino acids [33]. The results of the present study demonstrated that the combined application of N and Fe at different stages increases the relative content of essential amino acids in the *grape* peel. Application of N at different growth stages had a significant or strongly significant effect on the relative content of the 11 amino acids detected in *grape* peel; additionally, Fe application at these different stages also significantly affected the relative contents of L-Threonine, L-Lysine, L-Methionine, L(+)-Arginine, Glycine, and L-Glutamic acid. 

Flavonoids and non-flavonoid phenols are among the most important secondary metabolites in *grapes* [34]. Flavonoids are the most abundant phenolic substances in *grapes* and wines and are primarily distributed in the berry peel and seed coat; these components of *grapes* can resist the damage of ultraviolet rays and pathogens and have several functions, such as antioxidant activity [4]. Álvarez-Fernández et al. [35] found that Fe can improve photosynthetic efficiency, which can then affect the way vines use precursors to synthesize phenolic compounds or other secondary metabolites. Further, the present study demonstrated that the synergistic application of N and Fe at different growth stages affected the flavonoids in *grape* peel; additionally, the period in which N application occurred had a significant impact on these *grape* peel flavonoids. Compared with the N application at the early stage of the veraison period, the N application at the late stage of the veraison period can significantly improve the flavonoids in *grape* peel because the late veraison stage is the most active stage of flavonoid synthesis in *grapes*; therefore, rapid foliar N supplementation at the correct growth period can effectively improve flavonoids in *grapes* and wine [34]. 

## 4. Materials and Methods

### 4.1. Cultivation and Experimental Design 

The experiment was conducted from April 2022 to October 2022 at Lilan Winery (105°58′20″ E, 38°16′38″ N) in the core area of the wine *grape* production region of eastern Helan Mountain, China. The trial site is at an altitude of 1129 m, with sufficient light, an average annual temperature of 8.9 °C, an annual sunshine rate of >65%, an average annual precipitation of 190 mm, and a frost-free period of 180 days [30]. The soil type is light, gravelly live soil, and the soil texture is gravelly sandy soil. The *grapes* analyzed were 8-year-old Cabernet Sauvignon *grapes* planted in a north-south direction, with a “sloping frame” training system, plant spacing of 0.6 × 3.5 m, and plant density of 4760 plants per hectare. The trial was conducted with 4.5 tons of sheep manure per hectare as the base fertilizer; no chemical fertilizer was applied throughout the trial except for foliar N and Fe fertilization. The irrigation method used was drip irrigation with a fertility irrigation quota of 3000 m^3^ ha^−2^. Urea was used for foliar N application, and iron ethylenediaminetetraacetic acid (EDTA-Fe) was used for foliar Fe application. 

The chemical characteristics of the soil before the start of the experiment are presented in Table 6. Soil samples from different layers were collected using a soil auger. The following methods were used for analysis: alkaline hydrolysis diffusion for available nitrogen, molybdenum-antimony anti-spectrophotometry for available phosphorus, flame photometry for available potassium, Kjeldahl method for total nitrogen, DTPA extraction-Inductively Coupled Plasma Mass Spectrometry (ICP-MS, LC 1260 MS G6420A, Aglient, Thermofisher, Waltham, MA, USA) for available iron, and strong acid digestion-ICP-MS for total iron [36,37].

The experiment adopted a split-zone design (Table 7), with the N fertilizer application period as the main zone. Three alternative N treatments were utilized at different stages: N1 (N application in the early stage of the expansion period), N2 (N application from the early stage of expansion to the late stage of the veraison period), and N3 (N application at the late stage of the veraison period). The Fe fertilization period was used as a secondary zone. The three corresponding treatments of Fe were as follows: Fe1 (Fe application in the early stage of the expansion period), Fe2 (Fe application from the early stage of expansion to the late stage of the veraison period), and Fe3 (Fe application in the late stage of the veraison period). N fertilizer was applied with a mass concentration of 2.5‰ urea; Fe fertilizer was applied with a mass concentration of 1.5‰ EDTA-Fe. Overall, there was a total of 9 treatments, each with 3 replicates and a total plot area of 567 m^2^, which were sprayed with an electric sprayer (20 L, backpack-type electric sprayer with a stirring function, Zhunongli); other cultivation measures were maintained as before. 

### 4.2. Photosynthetic Characteristics and N and Fe Contents of Leaves 

The photosynthetic characteristics of *grape* leaves in each treatment, including the net photosynthetic rate (Pn), stomatal conductance (Gs), transpiration rate (Tr), intercellular CO_2_ concentration (Ci), and water use efficiency (WUE), were measured at 8:00 a.m. on 24 August 2022 using the LI-6800 convenient photosynthetic measurement system (LI-COR, Lincoln, NE, USA). From each treatment group, ten randomly selected leaves of uniform length and height on the same side of each plant were quickly placed in a sampling box with dry ice and brought back to the laboratory to determine chlorophyll a and chlorophyll b content. The chlorophyll a and b were extracted using an 80% acetone solution, and their concentrations were measured using spectrophotometry at specific wavelengths (typically 663 nm for chlorophyll a and 645 nm for chlorophyll b) [38,39].

Leaves and petioles of the plants were collected at the ripening stage of the *grapes*, killed at 100 °C, and dried at 65 °C until constant weight. The leaves were crushed in a crusher (1500 g, RS-FS1612 crusher, Rongshida, Royalstar, Hefei, China), passed through a 0.25 mm sieve, mixed, and bagged (100 mm × 150 mm transparent sealable bag, Weiyu, Yueqia, Shanghai, China). The total N content of the plants was determined using the H_2_SO_4_-H_2_O_2_ digestion-Kjeldahl method. The Fe content of the plant samples was determined using a rigorous analytical procedure. Initially, the samples were incinerated in a muffle furnace at 550 °C to ensure complete combustion of organic matter. Subsequently, the residual ash was analyzed using an inductively coupled plasma optical emission spectrometer (ICP-MS, LC 1260 MS G6420A, Aglient, Thermofisher, Waltham, MA, USA) for precise quantification of the Fe content [40,41].

### 4.3. Berry Quality, Yield, and Amino Acid and Flavonoid Content 

*Grapes* from each plot were harvested during ripening for yield measurement; additionally, the plants were surveyed to obtain the yield per plant, which was converted to yield per hectare in each plot according to the planting density. Specifically, 100 wine *grapes* were randomly selected from each plot, and their weight was measured using an electronic balance (capacity of 200 g and an accuracy of 0.01 g, TD20002A, LICHEN, Li Chen Bunsey instrument, Shanghai, China); additionally, berries were randomly selected, and their diameters measured using Vernier calipers (BXGYBKC-220779, SYNTEK, Deqing Sheng Taixin, Huzhou, China). Finally, three bunches of *grapes* were randomly selected from each plot, and their lengths were measured using a measuring tape. The total soluble solids content (SSC), representing the concentration of dissolved sugars, was determined using a handheld refractometer (0–90% BRIX, handheld sugar meter, AIREP, AIREP instrument, Zibo, China), while the reducing sugars were quantified through the process of 3,5-dinitrosalicylic acid titration, and the titratable acidity (Tartaric acid) content (TAC) was determined by NaOH titration [42]. *Grape* berries were rapidly frozen with liquid nitrogen and then manually ground into powder using a mortar and pestle; 5 g of the powder was extracted with acidified methanol (Methanol acidified with 0.1% hydrochloric acid (*v/v*)) and then centrifuged three times (MK-20RB high and low-speed cryo-centrifuge, Michael, Michael Technology, Shenzhen, China), followed ultrasonication for 15 min at a frequency of 40 kHz and a temperature of 25 °C, was used to determine tannin, total phenol and anthocyanin content. Tannin content was determined using the Folin–Denis method, which involves reacting tannins with phosphotungstomolybdic acid and measuring the blue color formed at 760 nm using a spectrophotometer. The total phenol content was determined using the Folin–Ciocalteu method, where phenols react with the Folin–Ciocalteu reagent to produce a blue complex that can be quantified spectrophotometrically at 765 nm. The anthocyanin content was determined using the pH differential method, which measures the absorbance of anthocyanins at two different pH levels (typically pH 1.0 and pH 4.5) to account for structural changes and calculates the concentration based on the differential absorbance [43,44,45]. 

On 25 September 2022, 30 berries were randomly selected from each plot, fifteen of which were used for the determination of amino acids and the other fifteen for the determination of flavonoids. For the determination of amino acids [46], seeds were removed prior to grinding the fruits into powder under liquid N. The seeds were collected into a 50 mL bag. The powder was collected into 50 mL centrifuge tubes and centrifuged at 10,000 rpm for 10 min at the same temperature conditions after soaking in distilled water for 4 h at 4 °C. The berries’ juice was filtered through a 0.45 μm nylon membrane, and 100 μL of the filtered juice was mixed with 50 μL of internal standard solution and 400 μL of 0.1 mol L^−1^ HCl. The separation column used was the AJS-02 column (4.6 mm × 150 mm, 3 μm) at a temperature of 45 °C. The alternative part, the determination of flavonoids in *grape* peels, was rapidly frozen in liquid N and vacuum dried (53 L, LCDZF-6050AB vacuum drier, LICHEN, Li Chen Bunsey instrument, Shanghai, China). The dried berries were then ground into powder for 1.5 min at a frequency of 30 Hz using a grinder (MM400 mixer mill, Retsch GmbH, Haan, GER). The powder was extracted using a 70% methanol solution and was mixed six times in a vortex mixer for 30 s each time with a 30-min interval and left overnight at 4 °C. The remaining homogenate was centrifuged (MK-20RB high-low speed refrigerated centrifuge, Michael, Michael Technology, Shenzhen, China) at 12,000 rpm for 10 min, and the supernatant was aspirated and filtered through a 0.22 µm PTFE microporous membrane for ultra-performance liquid chromatography-tandem mass spectrometry (UPLC-MS/MS) analysis using a Xevo TQ-S system (Waters, Milford, MA, USA) [47]. 

### 4.4. Statistical Analysis

All data were analyzed using SPSS 19.0 (IBM Co., Armonk, NY, USA). The graph creation and correlation analysis were performed using Origin 2022 (Origin Lab Co., Northampton, MA, USA). The Tukey Honestly Significant Difference (HSD) test was employed to assess variations among all treatments (*p* ≤ 0.05), while a two-way ANOVA was utilized to elucidate the interaction effects between N and Fe (* *p* ≤ 0.05; ** *p* ≤ 0.01; *** *p* ≤ 0.001; NS nonsignificant differences).

## 5. Conclusions

In this study, we determined that spraying N and Fe fertilizers at different growth stages affected the berry composition and quality of wine *grapes*. N application during the late stage of the veraison period and Fe application during the early stage of the berry expansion period significantly improved several parameters, such as increasing the photosynthetic parameters and soluble solid content of *grape* berries, decreasing the titratable acidity content, balancing the sugar-to-acid ratio, increasing the berry weight and yield, and increasing the relative content of certain amino acids and flavonoid compounds in berries. These results provide an important theoretical basis for local *grape* quality improvement and wine fermentation; however, the specific amino aci d and flavonoid metabolic pathways in berries during the *grape* veraison period remains unclear, and the mechanism of the combined effect of N and Fe on berry composition needs to be further established.

## Figures and Tables

**Figure 1 plants-13-02203-f001:**
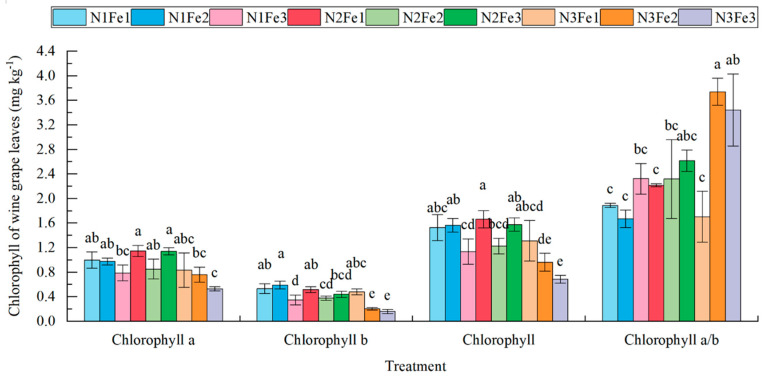
Chlorophyll content in wine grape leaves treated with co-applications of N and Fe. Effect of N and Fe co-application on the chlorophyll of wine grape leaves. Different lowercase letters (abcd) indicate significant differences (*p* < 0.05).

**Figure 2 plants-13-02203-f002:**
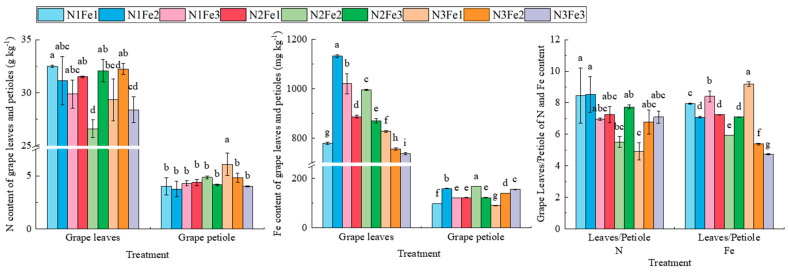
Effect of co-application of N, Fe, and leaves/petiole on N and Fe contents in leaves and petioles of wine grapes at different developmental stages. Different lowercase letters indicate significant differences (*p* < 0.05).

**Table 1 plants-13-02203-t001:** The effect of N and Fe application on photosynthetic parameters of wine grape leaves.

Treatment	Pn (μmol m^−2^ s^−1^)	Gs (mmol m^−2^ s^−1^)	Tr (vμmol m^−2^ s^−1^)	Ci (μmol mol^−1^)	WUE (%)
N1Fe1	14.20 a	124.5 a	3.73 a	167.5 e	3.81 bc
N1Fe2	11.5 b	89.50 b	3.00 bc	145.5 f	3.84 b
N1Fe3	8.95 cd	82.33 bc	3.00 bc	180.5 de	2.98 bcd
N2Fe1	9.43 c	123.3 a	3.37 ab	204.5 c	2.78 bcd
N2Fe2	8.1 de	76.50 c	2.70 c	188.0 d	2.91 bcd
N2Fe3	6.87 f	84.00 bc	2.96 bc	226.0 b	2.26 d
N3Fe1	13.35 a	122.0 a	2.11 d	332.0 a	6.48 a
N3Fe2	7.43 ef	65.00 d	1.76 d	179.0 de	2.72 cd
N3Fe3	7.47 ef	58.00 d	2.11 d	133.5 f	3.34 bcd
N application	**	**	**	**	**
Fe application	**	**	**	**	**
Interaction	**	**	NS	**	**

Pn—net photosynthetic rate; Gs—stomatal conductance; Tr—transpiration rate; Ci—intercellular CO2 concentration; WUE—water use efficiency; N1—nitrogen application in the early stage of expansion; N2—nitrogen application in the early stage of expansion to the later stage of veraison; N3—nitrogen application in the later stage of veraison; Fe1—iron application in the early stage of expansion; Fe2—iron application in the early stage of expansion to the later stage of veraison; Fe3—iron application in the later stage of veraison; Different letters indicate significant differences (*p* < 0.05); NS—no significant differences; **—significant at 1% levels.

**Table 2 plants-13-02203-t002:** The effect of N and Fe application on the morphological indices of wine grapes.

Treatment	Berry Size (mm)	Raceme Length (cm)	Berries Weight (g)	Yield Plant (kg)	Yield (kg ha^−2^)
N1Fe1	10.28 d	16.96 ab	101.4 c	1.15 b	5418 b
N1Fe2	11.44 bcd	15.28 b	127.4 abc	1.19 b	5626 b
N1Fe3	10.47 cd	15.77 b	127.9 abc	1.11 b	5227 b
N2Fe1	11.57 bcd	16.21 b	110.4 abc	1.16 b	5462 b
N2Fe2	13.21 ab	13.63 b	137.7 ab	1.18 b	5592 b
N2Fe3	12.13 bcd	15.88 b	102.7 bc	1.29 ab	6080 ab
N3Fe1	12.53 abc	21.29 a	143.1 a	1.56 a	7355 a
N3Fe2	14.55 a	17.74 ab	142.5 a	1.38 ab	6505 ab
N3Fe3	13.41 ab	15.68 b	139.7 a	1.28 ab	6064 ab
N application	**	**	**	NS	NS
Fe application	**	**	*	NS	NS
Interaction	NS	NS	*	NS	NS

N1—nitrogen application in the early stage of expansion; N2—nitrogen application in the early stage of expansion to the later stage of veraison; N3—nitrogen application in the later stage of veraison; Fe1—iron application in the early stage of expansion; Fe2—iron application in the early stage of expansion to the later stage of veraison; Fe3—iron application in the later stage of veraison; Different letters indicate significant differences (*p* < 0.05); NS—no significant differences; *, **—significant at 5% and 1% levels.

**Table 3 plants-13-02203-t003:** The effect of N and Fe co-application on wine grape quality.

Treatment	SSC (%)	TAC (Tartaric Acid %)	SSC/TAC (%)	Tannins (mg g^−1^)	Anthocyanins (mg g^−1^)	Total Phenols (mg g^−1^)
N1Fe1	24.40 ab	0.69 a	35.07 bc	3.92 bc	6.64 c	9.23 c
N1Fe2	22.97 ab	0.66 a	34.81 c	3.98 abc	9.05 b	11.46 ab
N1Fe3	24.17 ab	0.68 a	36.53 bc	4.23 ab	8.76 bc	11.36 ab
N2Fe1	21.80 b	0.60 ab	36.26 bc	4.16 ab	8.9 bc	10.92 b
N2Fe2	23.83 ab	0.58 ab	41.11 b	4.19 ab	6.49 c	9.26 c
N2Fe3	25.00 ab	0.66 a	37.83 bc	4.23 a	8.75 bc	12.63 a
N3Fe1	25.97 a	0.52 b	50.42 a	3.69 cd	7.28 bc	7.24 d
N3Fe2	24.13 ab	0.68 a	35.36 bc	3.73 cd	9.03 b	7.26 d
N3Fe3	25.27 ab	0.6 ab	41.92 b	3.58 d	12.56 a	7.68 d
N application	NS	*	*	**	NS	**
Fe application	NS	NS	NS	NS	*	**
Interaction	NS	NS	*	NS	*	**

SSC—soluble solid content; TAC—titratable acidity content; N1—nitrogen application in the early stage of expansion; N2—nitrogen application in the early stage of expansion to the later stage of veraison; N3—nitrogen application in the later stage of veraison; Fe1—iron application in the early stage of expansion; Fe2—iron application in the early stage of expansion to the later stage of veraison; Fe3—iron application in the later stage of veraison; Different letters indicate significant differences (*p* < 0.05); NS—no significant differences; *, **—significant at 5% and 1% levels.

**Table 4 plants-13-02203-t004:** Effect of N and Fe co-application on the relative content of essential amino acids.

Essential Amino Acid	N1Fe1	N1Fe2	N1Fe3	N2Fe1	N2Fe2	N2Fe3	N3Fe1	N3Fe2	N3Fe3	N Application	Fe Application	Interaction
	(mg L^−1^)											
L-Serine	1.17 e	1.15 e	2.47 de	5.05 bc	3.18 cde	2.53 de	7.57 a	6.59 ab	3.69 cd	**	NS	NS
L-Proline	0.82 e	0.73 e	2.22 bcd	2.91 abc	2.45 bcd	1.34 de	3.57 a	3.20 ab	1.68 cde	*	NS	**
L-Threonine	0.93 c	0.82 c	2.18 c	2.22 c	2.33 c	2.49 c	3.74 a	3.53 b	2.21 c	**	**	**
L-Aspartic Acid	0.60 c	0.62 c	1.46 bc	1.59 bc	1.48 bc	2.2 ab	3.42 a	1.85 bc	1.57 bc	**	NS	NS
L-Lysine	0.94 f	0.51 f	3.36 cd	4.52 c	1.33 ef	2.29 de	6.65 a	4.64 b	3.64 c	**	**	**
L-Methionine	0.46 c	0.21 c	1.53 c	1.52 c	1.85 c	0.89 c	2.88 b	2.97 a	0.79 c	*	**	**
L-Histidine	1.15 c	1.35 c	1.52 c	1.41 c	1.63 c	2.87 b	3.79 a	2.63 b	2.59 b	**	NS	**
L-Phenylalanine	0.98 c	0.77 c	2.43 c	2.84 bc	2.22 c	2.25 c	4.65 b	5.85 a	2.07 c	*	NS	**
L(+)-Arginine	1.07 b	1.57 b	1.17 b	1.90 b	1.75 b	1.97 b	2.99 a	2.77 a	2.09 b	**	**	**
Glycine	1.17 c	1.40 c	2.09 c	2.73 c	1.49 c	1.09 c	3.23 a	3.07 b	1.70 c	**	**	**
L-Glutamic acid	0.92 d	1.2 cd	1.71 bcd	2.33 b	0.99 d	1.69 bc	3.50 a	3.55 a	1.06 cd	**	*	**

N1—nitrogen application in the early stage of expansion; N2—nitrogen application in the early stage of expansion to the later stage of veraison; N3—nitrogen application in the later stage of veraison; Fe1—iron application in the early stage of expansion; Fe2—iron application in the early stage of expansion to the later stage of veraison; Fe3—iron application in the later stage of veraison; Different letters indicate significant differences (*p* < 0.05); NS—no significant differences; *, **—significant at 5% and 1% levels.

**Table 5 plants-13-02203-t005:** N and Fe co-application on flavonoids in wine grape peel.

Flavonoids	N1Fe1	N1Fe2	N1Fe3	N2Fe1	N2Fe2	N2Fe3	N3Fe1	N3Fe2	N3Fe3	N Application	Fe Application	Interaction
	(mg L^−1^)											
Silibinin	19.79 a	33.30 a	51.92 a	59.41 a	60.83 a	65.21 a	33.53 a	56.93 a	67.47 a	*	NS	NS
Puerarin	12.86 a	21.30 a	11.56 a	7.91 a	12.86 a	5.50 a	13.14 a	26.84 a	17.84 a	NS	NS	NS
Quercetin	678.7 ab	520.2 b	525.7 b	626.8 b	723.3 ab	698.3 ab	541.8 b	997.0 a	777.0 ab	**	NS	**
Genistein	10.89 ab	22.16 ab	11.16 ab	10.51 b	15.70 ab	7.36 b	58.94 a	24.59 ab	8.51 b	NS	NS	NS
Apigenin	3.44 b	4.09 b	4.86 b	2.73 b	2.04 b	3.21 b	64.98 a	4.31 b	2.86 b	*	*	**
Baicalein	212.9 a	335.1 a	285.5 a	174.4 a	241.8 a	231.9 a	371.2 a	469.2 a	260.1 a	NS	NS	NS
Baicalin	12.93 b	61.76 b	16.26 b	21.39 b	15.92 b	21.96 b	794.6 a	27.98 b	12.68 b	*	*	**
Rutin	2704 a	4565 a	2680 a	4138 a	1823 a	1987 a	3534 a	3896 a	3805 a	NS	NS	NS
hesperetin	16.6 b	59.00.b	33.70 b	20.40 b	21.30 b	17.4 b	379.9 a	24.00 b	5.90 b	*	*	**
Hesperidin	2354 b	3487 b	3238 b	2809 b	2373 b	2904 b	11313 a	3657 b	2962 b	**	*	**
Cianidanol	532.3 a	808.9 a	478.4 a	301.0 a	113.3 a	75.90 a	242.7 a	509.4 a	322.1 a	NS	NS	NS
Protocatechua-ldehyde	135.4 ab	198.1 ab	74.70 b	123.8 ab	137.5 ab	95.00 b	415.2 a	226.4 ab	140.8 ab	*	NS	NS
3,4-Dihydroxy-benzoic acid	624.2 a	663.3 a	545.3 a	464.1 a	721.4 a	543.1 a	957.3 a	1182.6 a	735.4 a	NS	NS	NS
Naringenin	16.34 a	31.29 a	50.60 a	33.19 a	26.21 a	33.71 a	63.47 a	52.15 a	49.41 a	*		
Luteolin	8.60 b	14.30 b	17.00 b	11.20 b	8.80 b	12.40 b	226.4 a	17.90 b	13.00 b	*	*	**
Myricetin	1488 a	1155 a	2074 a	1775 a	1351 a	830.0 a	1464 a	1513 a	2192 a	NS	NS	NS
Diosmin	3287 ab	3900 ab	4266 ab	3559 ab	2847 b	4103 ab	5491 a	4296 ab	3888 ab	NS	NS	NS
morin	137.7 a	179.0 a	117.8 a	96.60 a	165.0 a	170.9 a	219.7 a	1145.5 a	100.7 a	NS	NS	NS
Neohesperidin	4980 b	678.0 b	643.0 b	490.0 b	388.0 b	505.0 b	2541 a	721.0 b	571 b	**	*	**
Myricitrin	9170 a	7607 b	6245 b	6391 b	6165 b	6263 b	7238 b	7255 b	6228 b	**	***	***
Hyperoside	8200 a	4403 b	3789 b	3860 b	3784 b	3800 b	4148 b	5005 b	4075 b	***	***	***
Taxifolin 3-o-rhamnoside	160.1 bc	105.0 bc	91.19 c	92.76 c	96.74 c	153.0 bc	81.62 c	272.7 a	202.5 ab	***	*	***
Daidzin	28.44 a	71.62 a	38.91 a	39.03 a	41.88 a	33.84 a	28.40 a	54.15 a	36.40 a	NS	NS	NS
quercitrin	2050 b	14166 a	8784 ab	11246 ab	5551 ab	5264 ab	8909 ab	7486 ab	8891 ab	NS	NS	*
Vitexin	3.48 a	4.84 a	5.25 a	11.26 a	5.41 a	12.83 a	24.30 a	16.28 a	5.19 a	NS	NS	NS
artemisinin	1660 b	1785 b	1462 b	407.0 b	180.0 b	493.0 b	19882 a	490.0 b	215.0 b	*	*	**
astragalin	3050 a	10741 b	5986 b	7443 b	8353 b	5837 b	9405 b	5948 b	6017 b		*	***
Icariin	55.54 a	62.60 a	56.17 a	33.74 a	33.63 a	101.25 a	37.87 a	26.14 a	34.87 a	NS	NS	NS
Troxerutin	64.50 a	91.80 a	120.2 a	114.7 a	166.6 a	116.9 a	222.6 a	220.1 a	97.20 a	NS	NS	NS
L-Epicatechin	71.60 a	120.9 a	118.8 a	220.1 a	153.5 a	149.1 a	127.2 a	130.5 a	91.70 a	NS	NS	NS
Isorhamnetin	749.1 a	639.2 a	699.6 a	541.9 a	525.9 a	519.8 a	851.2 a	725.5 a	781.3 a	*	NS	NS
Genistin	605.0 a	910.0 a	851.0 a	790.0 a	721.0 a	991.0 a	1138 a	855.0 a	642.0 a	NS	NS	NS
procyanidin B2	34.65 a	44.41 a	40.76 a	32.28 a	23.60 a	20.41 a	27.51 a	55.81 a	44.04 a	NS	NS	NS

N1—nitrogen application in the early stage of expansion; N2—nitrogen application in the early stage of expansion to the later stage of veraison; N3—nitrogen application in the later stage of veraison; Fe1—iron application in the early stage of expansion; Fe2—iron application in the early stage of expansion to the later stage of veraison; Fe3—iron application in the later stage of veraison; Different letters indicate significant differences (*p* < 0.05); NS—no significant differences; *, **, ***—significant at 5%, 1% and 1‰ levels.

**Table 6 plants-13-02203-t006:** Physical and chemical properties of vineyard soil.

Soil Layer	pH	Organic Matter	Available N	Available P	Available K	Total N	Available Fe	Total Fe
(cm)		(g kg^−1^)	(mg kg^−1^)	(mg kg^−1^)	(mg kg^−1^)	(g kg^−1^)	(mg kg^−1^)	(mg kg^−1^)
0–20	8.32	6.26	33.42	7.30	145.0	0.58	4.90	16.62
20–40	8.47	5.78	25.07	6.98	81.11	0.25	3.90	16.03
40–60	8.40	4.82	18.67	7.07	72.13	0.23	3.40	15.12

**Table 7 plants-13-02203-t007:** N and Fe fertilizer application time.

Treatment	N Application Period	Specific Date	Fe Application Period	Specific Date
N1Fe1	Early stage of expansion period	9/6, 18/6, 29/6	Early stage of expansion period	9/6, 18/6, 29/6
N1Fe2	Early stage of expansion period	9/6, 18/6, 29/6	Expansion period to Veraison period	9/7, 16/7, 24/7
N1Fe3	Early stage of expansion period	9/6, 18/6, 29/6	Later stage of Veraison period	31/7, 6/8, 13/8
N2Fe1	Expansion period to Veraison period	9/7, 16/7, 24/7	Early stage of expansion period	9/6, 18/6, 29/6
N2Fe2	Expansion period to Veraison period	9/7, 16/7, 24/7	Expansion period to Veraison period	9/7, 16/7, 24/7
N2Fe3	Expansion period to Veraison period	9/7, 16/7, 24/7	Later stage of Veraison period	31/7, 6/8, 13/8
N3Fe1	Later stage of Veraison period	31/7, 6/8, 13/8	Early stage of expansion period	9/6, 18/6, 29/6
N3Fe2	Later stage of Veraison period	31/7, 6/8, 13/8	Expansion period to Veraison period	9/7, 16/7, 24/7
N3Fe3	Later stage of Veraison period	31/7, 6/8, 13/8	Later stage of Veraison period	31/7, 6/8, 13/8

## Data Availability

The data supporting the findings of this study are available from the corresponding author upon reasonable request.

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
