# Peer review of "Foliar Co-Applications of Nitrogen and Iron on Vines at Different Developmental Stages Impacts Wine Grape (Vitis vinifera L.) Composition"

_plants, 2024, doi:10.3390/plants13162203_

Round 1

Reviewer 1 Report

Comments and Suggestions for Authors

Major comments

1)lines 92-94 Over all, total leaf chlorophyll was highest under N2Fe1 treatment, at 1.66 mg g-1, which increased by 5.73-144.12% compared to the other treatments, with an overall performance as follows: N2Fe1>N2Fe3>N1Fe2>N1Fe1>N3Fe1>N2Fe2>N1Fe3>N3Fe2>N3Fe3’.The differences in most cases are statistically insignificant according to the presented data

2) Fe leaf/petiole distribution is rather attractive. In general, not absolute values are significant but ratios often give higher information. May be it will be interesting to evaluate leaf/petiole ratios for Fe (or N)?

4) no data of sugar content and method of their determination

3) indicate the units of Titratable Acidity- % per citric of malic acid?

 6) did the treatments change the phenolic distribution between peel and pulp?

7) tannins and anthocyanins belong to polyphenols. why the amount of tannins in Table 3 is higher than the total phenolics?

8) indicate the variations in chlorophyll a/b ratio according to different treatments

9) I should recommend to revise Material and Methods section and give more detailed description on the methods of determination of (i) chlorophyll (no data at all), (ii) phenolics including anthocyanines, tannins, (iii) sugar (no data) and indicate the method of available N, P, K, Fe in soil

Minor comments

1)Table 1- decipher all abbreviations in the footnote

2) Figure 1. ‘N and Fe co-application on the chlorophyll of wine grape leaves’ change to :  Effect of N and Fe co-application on the chlorophyll of wine grape leaves

3) Fig.2- add (A) and (B) to the title

4) Table 5 decipher ‘***’

5) line 233 ‘Soluble solids and titratable acids are the raw materials required for the synthesis of most organic acids in grapes’- but in general soluble solids is a sum of sugar, minerals and acids- and titratable acids refer to organic  acids- TA- is a sum of organic acids but not raw materials required for their synthesis

6)line 340 ‘the Fe content of the plants was burned in a muffle furnace at 550 °C and determined using an inductively coupled plasma optical emission spectrometer’- style revise

7) Reference list- use journals’ abbreviations. See author’s guidelines

8) ref 13,19,33, etc =use Italics for plant species

9) ref 32- add the number of the article

10) Tables 4,5- add units of the parameters

Author Response

Response to Reviewer 1 Comments

1. Summary

2. Questions for General Evaluation

Reviewer’s Evaluation

Response and Revisions

Does the introduction provide sufficient background and include all relevant references?

Yes

Is the research design appropriate?

Yes

Are the methods adequately described?

Must be improved

Are the results clearly presented?

Can be improved

Are the conclusions supported by the results?

Yes

3. Point-by-point response to Comments and Suggestions for Authors

Major comments

Comments 1: lines 92-94 Over all, total leaf chlorophyll was highest under N2Fe1 treatment, at 1.66 mg g-1, which increased by 5.73-144.12% compared to the other treatments, with an overall performance as follows: N2Fe1>N2Fe3>N1Fe2>N1Fe1>N3Fe1>N2Fe2>N1Fe3>N3Fe2>N3Fe3’.The differences in most cases are statistically insignificant according to the presented data.

Response 1: We appreciate the reviewer’s insightful comment. Upon careful review, we identified an inaccuracy in our initial presentation. We have revised the relevant section (lines 105-110) to accurately reflect the data. The revised text now reads:

“Overall, compared to the other treatments, the total chlorophyll content of the leaves under the N2Fe1 treatment was 1.66 mg g-1. This value was higher than that of N1Fe3, N2Fe2, N3Fe2, and N3Fe3 treatments by 8.17-52.29%, 24.64%-28.21%, 23.63%-57.24%, and 48.29%-65.56%, respectively.”

This adjustment ensures clarity and precision in reporting the chlorophyll content differences among the treatments. Thank you for highlighting this issue.

Comments 2: Fe leaf/petiole distribution is rather attractive. In general, not absolute values are significant but ratios often give higher information. May be it will be interesting to evaluate leaf/petiole ratios for Fe (or N)?

Response 2: We appreciate the reviewer's valuable suggestion. We have reprocessed the data to evaluate the leaf/petiole ratios for Fe and N content. Additionally, we have included relevant plots and have provided a thorough analysis and discussion in the “3. Discussion” section. This enhancement adds a deeper understanding of the nutrient distribution patterns. Thank you for this insightful recommendation.

Comments 3: no data of sugar content and method of their determination

Response 3: Thank you for your valuable feedback.  In this study, we have used soluble solids content (SSC) as a representative measure for sugar content.  To enhance clarity and consistency, we have standardized the term “sugar content” to “soluble solids content.”  We would like to provide the following explanation for the relationship between sugar content and SSC:

1.     Measurement Techniques: SSC is commonly measured using refractometers or density meters, which evaluate the refractive index or density of the solution.  Due to their high solubility and concentration, sugars have a significant impact on these measurements.

2.     Prevalence of Sugars in Fruits and Vegetables: In many fruits and vegetables, sugars such as glucose, fructose, and sucrose are the predominant components of soluble solids.  These sugars are synthesized through photosynthesis and are primarily responsible for the sweetness of the fruit.

3.     High Solubility of Sugars: Sugars exhibit a high solubility in water, dissolving completely to form a homogeneous solution.  As a result, sugars are the principal contributors to the soluble solids content measured.

We appreciate your understanding and hope this explanation clarifies the rationale behind our methodology.

Comments 4:indicate the units of Titratable Acidity- % per citric of malic acid?

Response 4: Thank you for your pertinent comment. When determining the titratable acidity of wine grape berries using the sodium hydroxide titration method, the primary acid measured is tartaric acid, which is the predominant organic acid in grapes. Additionally, grapes contain significant amounts of malic acid and minor quantities of citric acid. However, during the titration process, tartaric acid typically constitutes the majority of the titratable acidity, making it the principal acid considered.

To provide a more precise representation, we have incorporated detailed information about the titratable acids into the “4. Materials and Methods” section and updated Table 3 accordingly. This clarification should address the unit specification effectively.

Comments 5: did the treatments change the phenolic distribution between peel and pulp?

Response 5: Thank you for your insightful question. In this experiment, the grape berries were rapidly frozen in liquid nitrogen, then crushed whole in the laboratory and weighed as a powder. Due to this methodology, we did not separate the peel and pulp before analysis, and therefore, we do not have specific data on whether the treatments altered the phenolic distribution between these parts. Future studies could be designed to address this aspect more specifically by analyzing the peel and pulp separately.

Comments 6: tannins and anthocyanins belong to polyphenols. why the amount of tannins in Table 3 is higher than the total phenolics?

Response 6: Thank you for your insightful observation. The reason for this may be because the reagents used for the detection of tannins in the Folin-Denis method may react more strongly or more specifically with tannin compounds, resulting in higher apparent concentrations. In contrast, the Folin-Ciocalteu reagent, although used for the detection of a wide range of phenolics, may not react as strongly with tannins, resulting in lower levels of tannins in the total phenolics. The Folin-Denis method measures the blue color that is formed when tannic acid reacts with phosphotungstic molybdic acid. This method is more sensitive or gives a higher reading for tannins. The Folin-Ciocalteu method measures the blue color complex formed when phenols react with Folin-Ciocalteu reagent. This method covers a wide range of phenolic compounds and may dilute the apparent concentration of tannins in the total phenolic content. ​We have corrected the data and reanalyzed them to ensure accurate representation.

Comments 7: indicate the variations in chlorophyll a/b ratio according to different treatments

Response 7: Thank you for your valuable comment. We have reprocessed the data to evaluate the variations in the chlorophyll a/b ratio under different treatments. Additionally, we have included a figure illustrating these variations and provided a detailed analysis and discussion in the “3. Discussion” section. This enhancement offers a clearer understanding of the impact of the treatments on the chlorophyll a/b ratio.

Comments 8: I should recommend to revise Material and Methods section and give more detailed description on the methods of determination of (i) chlorophyll (no data at all), (ii) phenolics including anthocyanines, tannins, (iii) sugar (no data) and indicate the method of available N, P, K, Fe in soil

Response 8: Thank you for your insightful suggestions. We have revised the Materials and Methods section to include detailed descriptions of the methods used for determining chlorophyll, phenolics (including anthocyanins and tannins), and soil nutrients (N, P, K, Fe).  Additionally, we have provided an explanation of the sugar determination method in our response to Comment 3.

Minor comments

Comments 1: Table 1- decipher all abbreviations in the footnote

Response 1: Thank you for your valuable feedback. We have now added footnotes to Table 1 in line 82 of the manuscript to clarify the abbreviations Pn, Gs, Tr, Ci, and WUE. We appreciate your attention to detail and hope this improves the clarity of our table.

Comments 2: Figure 1.‘N and Fe co-application on the chlorophyll of wine grape leaves’ change to : ‘Effect of N and Fe co-application on the chlorophyll of wine grape leaves

Response 2: Agree, thank you for your suggestion. I have made the change as requested.

Comments 3: Fig.2- add (A) and (B) to the title

Response 3: Thank you for your insightful suggestion. I have added (A) and (B) to the title of Figure 2 as requested.

Comments 4: Table 5 decipher ‘***’

Response 4: Thank you for pointing that out. I have deciphered the ‘***’ in Table 5 as requested.

Comments 5: line 233 ‘Soluble solids and titratable acids are the raw materials required for the synthesis of most organic acids in grapes’- but in general soluble solids is a sum of sugar, minerals and acids- and titratable acids refer to organic  acids- TA- is a sum of organic acids but not raw materials required for their synthesis

Response 5: Thank you for pointing out this inaccuracy. The sentence has been corrected to accurately reflect the composition and role of soluble solids and titratable acids. The revised statement can be found in lines 266-267.

Comments 6: line 340 ‘the Fe content of the plants was burned in a muffle furnace at 550 °C and determined using an inductively coupled plasma optical emission spectrometer’- style revise

Response 6: We appreciate your suggestion.  The sentence has been revised for clarity and precision.  The updated sentence can be found in lines 386-387.

Comments 7: Reference list- use journals’ abbreviations. See author’s guidelines

Response 7: Thank you for your comment. We have revised the reference list to use the appropriate journal abbreviations in accordance with the author’s guidelines.

Comments 8: ref 13,19,33, etc =use Italics for plant species

Response 8: Thank you for your comment. We have revised the references to use italics for plant species as required.

Comments 9: ref 32- add the number of the article

Response 9: Thank you for your comment.  We have updated reference 32 to include the number of the article as required.

Comments 10: Tables 4,5- add units of the parameters

Response 10: Thank you for your valuable suggestion. I have added the units of the parameters in Tables 4 and 5 as requested.

4. Response to Comments on the Quality of English Language

Response: We have revised Quality of English Language for the full manuscript to ensure better compliance with the journal's requirements.

5. Additional clarifications

Response to Reviewer 2 Comments

1. Summary

2. Questions for General Evaluation

Reviewer’s Evaluation

Response and Revisions

Does the introduction provide sufficient background and include all relevant references?

Yes

Is the research design appropriate?

Yes

Are the methods adequately described?

Must be improved

Are the results clearly presented?

Must be improved

Are the conclusions supported by the results?

Yes

3. Point-by-point response to Comments and Suggestions for Authors

Comments 1: Change the title "Effects of N and Fe co-application at different growth stages on grape berry composition" to "Foliar co-applications of nitrogen and iron on vines at different developmental stages impacts wine grape composition."

Response 1: Thank you for pointing this out. I agree with your suggestion and have accordingly changed the title on page 1, lines 2 and 3 of the manuscript.

Comments 2: Change "The co-application of N and Fe can promote the improvement of berry composition and formation of flavor compounds in wine grapes" to "The co-application of N and Fe can improve wine grape composition and promote the formation of flavor compounds."

Response 2: I appreciate your suggestion. I have revised the manuscript to reflect this change, updating the sentence on lines 8 and 9 as recommended.

Comments 3: This sentence as is seems incmplete. Combine with the following sentence. For example: To understand the effects of foliar co-application of N and Fe on wine grape quality and flavonoid content, urea and EDTA-FE were sprayed at three different developmental stages.

Response 3: Thank you for your observation. I have revised the manuscript to combine the sentences as suggested, ensuring a clearer and more complete statement.

Comments 4: The text on lines 13-16 is not clear. What were the most important results? Do you mean that berries treated at pre-expansion and late veraison stages had higher SSC, etc.? What were the best N+Fe treatments?

Response 4: Thank you for your comment. We have rewritten the passage to more clearly express the most important results. The revised text can be found in lines 15-19.

Comments 5: Start the sentence with grapes. For example: Grapes (Vitis vinifera L.) are one of the most important fruit crops grown worldwide because of their yield and economic value. Latin names should always be in italic. The introduction can be improved by re-organizing the content. Besides, you need to provide references to support your statements. For example:

1.Importance of wine grapes worldwide and in China

2.Growing conditions and fertilization

3.Impact of agricultural practices (fertilization) on grape quality

4.Specifics about N and Fe

5.Importance of your study (what makes it different from other studies)

6.Aims of your study

Response 5: Thank you for your detailed and constructive feedback. Based on your comments, we have completely revised the “Introduction” section. We have added the following sections: “Impact of agricultural practices (fertilization) on grape quality”, “Importance of your study”, and “Aims of your study”. Additionally, we ensured that Latin names are italicized and provided references to support our statements. The revised content has been reorganized to improve clarity and coherence.

Comments 6: “Enhances the volatility of aromatic compounds” - What does this mean?

Response 6: Thank you for your valuable feedback. High sugar accumulation in berries enhances the volatility of aromatic compounds. This means that as the sugar levels in the berries increase, it facilitates the release and perception of these aromatic compounds. The sugars interact with these compounds, making them more prone to evaporation and, therefore, more detectable in the wine's aroma. This interaction significantly enhances the aromatic profile of the wine, contributing to a richer and more complex bouquet. I appreciate your attention to this detail and have revised the manuscript accordingly.

Comments 7: "These secondary metabolites have antioxidant functions and can resist the damage caused by ultraviolet radiation and grape pathogens." Not clear. Do you mean that polyphenols (secondary metabolites) can protect the plant against UV radiation and pathogen attack?

Response 7: Yes, polyphenols, which are secondary metabolites, have antioxidant properties that protect the plant against damage caused by ultraviolet (UV) radiation and pathogen attack. These compounds help to neutralize harmful free radicals generated by UV exposure and can inhibit the growth of various grape pathogens, thereby enhancing the plant's resilience and overall health.

Comments 8: Change “Effect of N and Fe co-application at different stages on physiological growth indexes of wine grape leaves” to “Effect of co-application of N and Fe on photosynthetic parameters of wine grape leaves at different developmental stages.”

Response 8: I appreciate your suggestion. The title has been updated to “Effect of co-application of N and Fe on photosynthetic parameters of wine grape leaves at different developmental stages” on lines 69 and 70 of the manuscript.

Comments 9: Why separate N from Fe treatments when you have them together on the table? Can you, for example, say: "The Pn content of grape leaves from treatments N1Fe1 and N3Fe1 was significantly higher than that from other treatments, whereas Gs content was higher in leaves from N1Fe1, N2Fe1, and N3Fe2 than in other treatments..." or "Grape leaves from treatment N1Fe1 had the highest content in Pn, Gs, and Tr. Leaves from treatment N3Fe1 had the highest content in Pn, Gs, Tr, and WUE..."

Response 9: Thank you for your insightful comment. Based on your suggestions, we have completely revised this part of the description to more effectively convey the combined effects of N and Fe treatments.

Comments 10: Include in the table1 footnote what each of these acronyms mean.

Response 10: Thank you for pointing that out. I have added the definitions of each acronym to the footnote of Table 1 to ensure clarity.

Comments 11: Change “N and Fe co-application on the chlorophyll of wine grape leaves” to “Chlorophyll content in wine grape leaves treated with co-applications of N and Fe.”

Response 11: Thank you for your input. I have updated the text on line 96 to “Chlorophyll content in wine grape leaves treated with co-applications of N and Fe” as suggested.

Comments 12: Change “Effect of N and Fe co-application at different stages on N and Fe content in leaves and petioles of wine grape” to “Effect of co-application of N and Fe on N and Fe contents in leaves and petioles of wine grape at different developmental stages.”

Response 12: I appreciate your suggestion. The text on line 97 has been revised to “Effect of co-application of N and Fe on N and Fe contents in leaves and petioles of wine grape at different developmental stages” as requested.

Comments 13: Change “The effect of N and Fe synergism on N and Fe content in leaves and petioles of wine grape” to “Effect of co-application of N and Fe on N and Fe contents in leaves and petioles of wine grape at different developmental stages.”

Response 13: Thank you for your recommendation. I have updated the text on line 113 to “Effect of co-application of N and Fe on N and Fe contents in leaves and petioles of wine grape at different developmental stages.”

Comments 14: Change “Effect of N and Fe co-application at different stages on morphological indicators and yield of wine grape” to “Effect of co-application of N and Fe on N and Fe contents in leaves and petioles of wine grape at different developmental stages.”

Response 14: I appreciate your feedback. The text on line 115 has been revised to “Effect of co-application of N and Fe on N and Fe contents in leaves and petioles of wine grape at different developmental stages” as suggested.

Comments 15: On line 118, "indices" - Be consistent, use "indices" or "indicators," not both.

Response 15: Thank you for your observation. I have made the necessary changes to ensure consistency, using "indices" throughout the manuscript.

Comments 16: On line 123, the "highest" - Do you mean that the morphological indicators were the highest?

Response 16: Thank you for highlighting this issue.   There was a problem with the phrasing of this sentence, and we have revised this part of the description to make it clearer.   The updated content can be found in lines 143-145.

Comments 17: In Table 2, "100-berries weight" - If you explain in Materials and Methods (M&M) that the number of berries used in the sample was 100 (n=100), there is no need to indicate it here. You can also add this information to the figure footnote.

Response 17: Thank you for your suggestion. I have clarified in the Materials and Methods section that the number of berries used in the sample was 100 (n=100) and added this information to the figure footnote. Consequently, I have removed the indication from Table 2.

Comments 18: In Table 2, does "Yieled" refer to Fruit?

Response 18: Yes, "Yield" refers to grape fruit.

Comments 19: Change “Effect of N and Fe co-application at different stages on wine grape quality” to “Effect of co-application of N and Fe on wine grape quality at different developmental stages.”

Response 19: Thank you for your suggestion. I have revised the text on line 135 to “Effect of co-application of N and Fe on wine grape quality at different developmental stages” as recommended.

Comments 20: On lines 136-138, please rewrite. For example: "The SSC of wine grapes from the N3Fe1 treatment was 2.78 to 19.13% higher than the other treatments (Table 3)."

Response 20: Thank you for your suggestion.  We have modified this sentence to read: “The SSC of wine grapes from the N3Fe1 treatment was 2.78 to 19.13% higher than the other treatments (Table 3).” The revised text can be found in lines 155-156.

Comments 21: On line 139, "titratable acidity" - Do you mean: "The titratable acidity of wine grapes from the N3Fe1 treatment was 6.67 to 18.84% lower than the other treatments"?

Response 21: Yes, we have modified this sentence to read: “The titratable acidity of wine grapes from the N3Fe1 treatment was 6.67 to 18.84% lower than the other treatments.” The revised text can be found in lines 156-157.

Comments 22: On lines 141-150, please rewrite (see suggestion above).

Response 22: Thank you for your suggestion. We have rewritten this section to make it clearer. The revised content can be found in lines 157-164.

Comments 23: Change “application” to “co-application” on line 151.

Response 23: I appreciate your input. The term on line 151 has been updated to “co-application” as suggested.

Comments 24: In Table 3, change “Soluble solids” to “Soluble solids content” or “SSC.”

Response 24: Thank you for your suggestion. We have made the change to "Soluble solids content" in Table 3.

Comments 25: Change “Titrate acid” to “Titratable acidity” in Table 3.

Response 25: Thank you for pointing this out. We have changed the reference to “Titrate acid” in Table 3.

Comments 26: Change “Effect of N and Fe co-application at different stages on relative content of essential amino acids in wine grapes” to “Effect of co-application of N and Fe on wine grape relative content of essential amino acids at different developmental stages.”

Response 26: Thank you for your suggestion. I have updated the text on line 156 to “Effect of co-application of N and Fe on wine grape relative content of essential amino acids at different developmental stages.”

Comments 27: Change “significantly increased the” to “showed a significantly higher content” on line 160.

Response 27: I appreciate your input. The text on line 160 has been revised to “showed a significantly higher content” as recommended.

Comments 28: Change “Effect of N and Fe co-application at different stages on flavonoids in wine grape pericarp” to “Effect of co-application of N and Fe on wine grape yield flavonoids content at different developmental stages.”

Response 28: Thank you for your recommendation. I have revised the text on line 176 to “Effect of co-application of N and Fe on wine grape yield flavonoids content at different developmental stages” as suggested.

Comments 29: On line 178, what do you mean by "flavonoid species" and "flavonoid substances"? Compounds?

Response 29: Thank you for pointing this out. The terms "flavonoid species" and "flavonoid substances" refer to different compounds within the flavonoid family. We acknowledge that the initial presentation was unclear and have standardized the nomenclature.

Flavonoid species: This term refers to the specific types or varieties of flavonoid compounds detected in the wine grape peel. For instance, the analysis identified 33 different flavonoid compounds out of a total of 46 that were screened. Examples of these flavonoid species mentioned in the text include genistein, apigenin, baicalin, and hesperetin.

Flavonoid substances: This term is essentially synonymous with "flavonoid species" in this context and refers to the individual flavonoid compounds that were detected and quantified in the grape.

Comments 30: Change “N or Fe” to “co-applications of N and Fe” on line 206.

Response 30: I appreciate your input. The text on line 206 has been updated to “co-applications of N and Fe” as requested.

Comments 31: Change “Soluble solids and titratable acids” to “Sugars and organic acids” on line 233.

Response 31: Thank you for pointing that out. I have corrected “Soluble solids and titratable acids” to “Sugars and organic acids” on line 233.

Comments 32: Change “most organic acids in grapes” to “many other compounds” on line 234.

Response 32: I appreciate your feedback. The text on line 234 has been revised to “many other compounds” as suggested.

Comments 33: On lines 251-257, you should include references to previously published studies to support your assumptions and data.

Response 33: Thank you for your suggestion. We have supplemented references to previously published studies to support our assumptions and data in lines 251-257.

Comments 34: On lines 299-300, this needs a reference. Where did you obtain these values/information?

Response 34: Thank you for pointing this out. We have supplemented the necessary reference to support the values and information mentioned in lines 299-300.

Comments 35: On line 303, should “training” be “training or draining”?

Response 35: Thank you for your query. In the context of viticulture and grapevine management, the term "training system" is correct. It refers to the methods used to shape and support the grapevines for optimal growth, sunlight exposure, and ease of management. Training System: This involves various techniques and structures used to guide the growth of grapevines, ensuring that the canopy is well-managed and the fruit is exposed to adequate sunlight. Common training systems include the "VSP" (Vertical Shoot Positioning), "Guyot," "Pergola," and in this case, the "sloping frame" system. These systems help improve grape quality and facilitate vineyard management tasks like pruning, spraying, and harvesting.

Comments 36: On line 325, for “electric sprayer,” please provide specifications, name, manufacturer, etc.

Response 36: Thank you for your suggestion. The electric sprayer used is a 20L, backpack-type, electric sprayer with a stirring function, Zhunongli.

Comments 37: On line 333, for "selected leaves," please specify how many.

Response 37: Thank you for your suggestion. We have clarified the exact number of leaves and revised the sentence to: “From each treatment group, ten randomly selected leaves of uniform length and height on the same side of each plant were quickly placed in a sampling box with dry ice and brought back to the laboratory to determine chlorophyll a and chlorophyll b content.”

Comments 38: On line 338, for "crusher," please specify the type of crusher.

Response 38: I have specified the type of crusher used in the manuscript. It is a 1500g, RS-FS1612 crusher, Rongshida.

Comments 39: On line 339, for "bag ged," please specify the type of bag.

Response 39: To address your comment, the specific type of bag used in the study is a 100mm x 150mm transparent sealable bag, manufactured by Weihyu.

Comments 40: On line 340, "and the Fe content of the plants was burned" is not clear.

Response 40: Thank you for your feedback.  We have revised the paragraph for clarity.  The updated sentence now reads: “The Fe content of the plant samples was determined using a rigorous analytical procedure.  Initially, the samples were incinerated in a muffle furnace at 550°C to ensure complete combustion of organic matter.  Subsequently, the residual ash was analyzed using an inductively coupled plasma optical emission spectrometer (ICP-OES, Varian, USA) for precise quantification of the Fe content.”  This revised text can be found in lines 385-389.

Comments 41: On line 348, for "electronic balance," please provide specifics of the balance used.

Response 41: I appreciate your input. The electronic balance used in the study is the TD20002A model by LICHEN, with a capacity of 200g and an accuracy of 0.01g.

Comments 42: On line 349, For "Vernier calipers," please provide more specifications (manufacturer, location, etc.).

Response 42: I appreciate your attention to detail. The Vernier calipers used are model BXGYBKC-220779, manufactured by SYNTEK.

Comments 43: On line 352, for "handheld sugar meter," please provide more specifications.

Response 43: Thank you for pointing that out. The handheld sugar meter used is a range of 0-90% BRIX, manufactured by AIREP.

Comments 44: On line 353, for "grains," do you mean berries or seeds?

Response 44: Thank you for the clarification. We have modified "grains" to "berries".

Comments 45: On line 354, for "powdered," how were they powdered?

Response 45: Thank you for your question. The samples were flash-frozen using liquid nitrogen and then ground into powder.

Comments 46: On line 354, for "certain amount," how much precisely?

Response 46: Thank you for pointing this out. We have specified the amount as 5 grams. This detail has been added to section “4.3 Berry quality, yield, and amino acid and flavonoid content” on line 405.

Comments 47: On line 355, for "acidified methanol," please provide more details.

Response 47: Thank you for your suggestion. We have specified that the methanol was acidified with 0.1% hydrochloric acid (v/v). This information has been included in section “4.3 Berry quality, yield, and amino acid and flavonoid content” on line 406.

Comments 48: On line 355, for "centrifuged," please provide the type of centrifuge (model, manufacturer, etc.).

Response 48: I appreciate your attention to detail. The centrifuge used is the MK-20RB high-low speed refrigerated centrifuge, manufactured by Michael.

Comments 49: On line 355, for "several times," how many times?

Response 49: Thank you for your question. We have specified that the centrifugation was performed three times. This detail has been added to section “4.3 Berry quality, yield, and amino acid and flavonoid content” on line 408.

Comments 50: On line 356, for "ultrasonication," please provide more details.

Response 50: Thank you for your suggestion. We have specified that ultrasonication was performed for 15 minutes at a frequency of 40 kHz and a temperature of 25°C. This information has been added to section “4.3 Berry quality, yield, and amino acid and flavonoid content.”

Comments 51: On line 361, for "repeat picking twice," were the 30 berries harvested twice (2 x 30)? At different dates?

Response 51: Thank you for pointing out the lack of clarity. To clarify, a total of 30 berries were picked, with 15 used for the determination of amino acids and the remaining 15 used for the determination of flavonoids. We have rewritten this section to make it clearer. The revised text can be found in lines 419-421.

Comments 52: On line 364, for "soaking," soaking in what?

Response 52: Thank you for your question. We have specified that the berries were soaked in distilled water. This detail has been added to section “4.3 Berry quality, yield, and amino acid and flavonoid content” on lines 423-425.

Comments 53: On line 368, for "column," please provide more specifications.

Response 53: Thank you for your suggestion. We have specified that the column used was 4.6 mm × 150 mm with a particle size of 3 μm. This detail has been added to section “4.3 Berry quality, yield, and amino acid and flavonoid content” on line 428.

Comments 54: On line 370, for "dried," please provide specifications of the vacuum drier.

Response 54:Thank you for your observation. The samples were dried using a 53L LCDZF-6050AB vacuum drier, manufactured by LICHEN.

Comments 55: On line 371, for "MM400, Retsch," please provide specifications.

Response 55: Thank you for your suggestion. We have specified that the equipment used is the MM400 mixer mill from Retsch GmbH, Haan, Germany. This detail has been added to section “4.3 Berry quality, yield, and amino acid and flavonoid content” on line 432.

Comments 56: On line 374, for "centrifuged," please provide centrifuge specifications.

Response 56: Thank you for your suggestion. We have included the specifications for the centrifuge in section “4.3 Berry quality, yield, and amino acid and flavonoid content.”

Comments 57: On line 375, for "microporous membrane," please provide specifications.

Response 57: Thank you for your suggestion. We have specified that the microporous membrane used is a 0.22 µm PTFE membrane. This detail has been added to section “4.3 Berry quality, yield, and amino acid and flavonoid content” on line 437.

Comments 58: On line 376, for "UPLC-MS/MS," please provide specifications (model, manufacturer, location, etc.).

Response 58: Thank you for your suggestion. We have included the specifications for the UPLC-MS/MS, including the model and manufacturer, in section “4.3 Berry quality, yield, and amino acid and flavonoid content” on line 438.

4. Response to Comments on the Quality of English Language

Response: We have revised Quality of English Language for the full manuscript to ensure better compliance with the journal's requirements.

5. Additional clarifications

Reviewer 2 Report

Comments and Suggestions for Authors

Please see comments, suggestions and edits on the attached file.

Comments on the Quality of English Language

The English is acceptable but needs to be polished. I made some edits already. Please see file attached for further comments and suggestions.

Author Response

Response to Reviewer 2 Comments

1. Summary

2. Questions for General Evaluation

Reviewer’s Evaluation

Response and Revisions

Does the introduction provide sufficient background and include all relevant references?

Yes

Is the research design appropriate?

Yes

Are the methods adequately described?

Must be improved

Are the results clearly presented?

Must be improved

Are the conclusions supported by the results?

Yes

3. Point-by-point response to Comments and Suggestions for Authors

Comments 1: Change the title "Effects of N and Fe co-application at different growth stages on grape berry composition" to "Foliar co-applications of nitrogen and iron on vines at different developmental stages impacts wine grape composition."

Response 1: Thank you for pointing this out. I agree with your suggestion and have accordingly changed the title on page 1, lines 2 and 3 of the manuscript.

Comments 2: Change "The co-application of N and Fe can promote the improvement of berry composition and formation of flavor compounds in wine grapes" to "The co-application of N and Fe can improve wine grape composition and promote the formation of flavor compounds."

Response 2: I appreciate your suggestion. I have revised the manuscript to reflect this change, updating the sentence on lines 8 and 9 as recommended.

Comments 3: This sentence as is seems incmplete. Combine with the following sentence. For example: To understand the effects of foliar co-application of N and Fe on wine grape quality and flavonoid content, urea and EDTA-FE were sprayed at three different developmental stages.

Response 3: Thank you for your observation. I have revised the manuscript to combine the sentences as suggested, ensuring a clearer and more complete statement.

Comments 4: The text on lines 13-16 is not clear. What were the most important results? Do you mean that berries treated at pre-expansion and late veraison stages had higher SSC, etc.? What were the best N+Fe treatments?

Response 4: Thank you for your comment. We have rewritten the passage to more clearly express the most important results. The revised text can be found in lines 15-19.

Comments 5: Start the sentence with grapes. For example: Grapes (Vitis vinifera L.) are one of the most important fruit crops grown worldwide because of their yield and economic value. Latin names should always be in italic. The introduction can be improved by re-organizing the content. Besides, you need to provide references to support your statements. For example:

1.Importance of wine grapes worldwide and in China

2.Growing conditions and fertilization

3.Impact of agricultural practices (fertilization) on grape quality

4.Specifics about N and Fe

5.Importance of your study (what makes it different from other studies)

6.Aims of your study

Response 5: Thank you for your detailed and constructive feedback. Based on your comments, we have completely revised the “Introduction” section. We have added the following sections: “Impact of agricultural practices (fertilization) on grape quality”, “Importance of your study”, and “Aims of your study”. Additionally, we ensured that Latin names are italicized and provided references to support our statements. The revised content has been reorganized to improve clarity and coherence.

Comments 6: “Enhances the volatility of aromatic compounds” - What does this mean?

Response 6: Thank you for your valuable feedback. High sugar accumulation in berries enhances the volatility of aromatic compounds. This means that as the sugar levels in the berries increase, it facilitates the release and perception of these aromatic compounds. The sugars interact with these compounds, making them more prone to evaporation and, therefore, more detectable in the wine's aroma. This interaction significantly enhances the aromatic profile of the wine, contributing to a richer and more complex bouquet. I appreciate your attention to this detail and have revised the manuscript accordingly.

Comments 7: "These secondary metabolites have antioxidant functions and can resist the damage caused by ultraviolet radiation and grape pathogens." Not clear. Do you mean that polyphenols (secondary metabolites) can protect the plant against UV radiation and pathogen attack?

Response 7: Yes, polyphenols, which are secondary metabolites, have antioxidant properties that protect the plant against damage caused by ultraviolet (UV) radiation and pathogen attack. These compounds help to neutralize harmful free radicals generated by UV exposure and can inhibit the growth of various grape pathogens, thereby enhancing the plant's resilience and overall health.

Comments 8: Change “Effect of N and Fe co-application at different stages on physiological growth indexes of wine grape leaves” to “Effect of co-application of N and Fe on photosynthetic parameters of wine grape leaves at different developmental stages.”

Response 8: I appreciate your suggestion. The title has been updated to “Effect of co-application of N and Fe on photosynthetic parameters of wine grape leaves at different developmental stages” on lines 69 and 70 of the manuscript.

Comments 9: Why separate N from Fe treatments when you have them together on the table? Can you, for example, say: "The Pn content of grape leaves from treatments N1Fe1 and N3Fe1 was significantly higher than that from other treatments, whereas Gs content was higher in leaves from N1Fe1, N2Fe1, and N3Fe2 than in other treatments..." or "Grape leaves from treatment N1Fe1 had the highest content in Pn, Gs, and Tr. Leaves from treatment N3Fe1 had the highest content in Pn, Gs, Tr, and WUE..."

Response 9: Thank you for your insightful comment. Based on your suggestions, we have completely revised this part of the description to more effectively convey the combined effects of N and Fe treatments.

Comments 10: Include in the table1 footnote what each of these acronyms mean.

Response 10: Thank you for pointing that out. I have added the definitions of each acronym to the footnote of Table 1 to ensure clarity.

Comments 11: Change “N and Fe co-application on the chlorophyll of wine grape leaves” to “Chlorophyll content in wine grape leaves treated with co-applications of N and Fe.”

Response 11: Thank you for your input. I have updated the text on line 96 to “Chlorophyll content in wine grape leaves treated with co-applications of N and Fe” as suggested.

Comments 12: Change “Effect of N and Fe co-application at different stages on N and Fe content in leaves and petioles of wine grape” to “Effect of co-application of N and Fe on N and Fe contents in leaves and petioles of wine grape at different developmental stages.”

Response 12: I appreciate your suggestion. The text on line 97 has been revised to “Effect of co-application of N and Fe on N and Fe contents in leaves and petioles of wine grape at different developmental stages” as requested.

Comments 13: Change “The effect of N and Fe synergism on N and Fe content in leaves and petioles of wine grape” to “Effect of co-application of N and Fe on N and Fe contents in leaves and petioles of wine grape at different developmental stages.”

Response 13: Thank you for your recommendation. I have updated the text on line 113 to “Effect of co-application of N and Fe on N and Fe contents in leaves and petioles of wine grape at different developmental stages.”

Comments 14: Change “Effect of N and Fe co-application at different stages on morphological indicators and yield of wine grape” to “Effect of co-application of N and Fe on N and Fe contents in leaves and petioles of wine grape at different developmental stages.”

Response 14: I appreciate your feedback. The text on line 115 has been revised to “Effect of co-application of N and Fe on N and Fe contents in leaves and petioles of wine grape at different developmental stages” as suggested.

Comments 15: On line 118, "indices" - Be consistent, use "indices" or "indicators," not both.

Response 15: Thank you for your observation. I have made the necessary changes to ensure consistency, using "indices" throughout the manuscript.

Comments 16: On line 123, the "highest" - Do you mean that the morphological indicators were the highest?

Response 16: Thank you for highlighting this issue.   There was a problem with the phrasing of this sentence, and we have revised this part of the description to make it clearer.   The updated content can be found in lines 143-145.

Comments 17: In Table 2, "100-berries weight" - If you explain in Materials and Methods (M&M) that the number of berries used in the sample was 100 (n=100), there is no need to indicate it here. You can also add this information to the figure footnote.

Response 17: Thank you for your suggestion. I have clarified in the Materials and Methods section that the number of berries used in the sample was 100 (n=100) and added this information to the figure footnote. Consequently, I have removed the indication from Table 2.

Comments 18: In Table 2, does "Yieled" refer to Fruit?

Response 18: Yes, "Yield" refers to grape fruit.

Comments 19: Change “Effect of N and Fe co-application at different stages on wine grape quality” to “Effect of co-application of N and Fe on wine grape quality at different developmental stages.”

Response 19: Thank you for your suggestion. I have revised the text on line 135 to “Effect of co-application of N and Fe on wine grape quality at different developmental stages” as recommended.

Comments 20: On lines 136-138, please rewrite. For example: "The SSC of wine grapes from the N3Fe1 treatment was 2.78 to 19.13% higher than the other treatments (Table 3)."

Response 20: Thank you for your suggestion.  We have modified this sentence to read: “The SSC of wine grapes from the N3Fe1 treatment was 2.78 to 19.13% higher than the other treatments (Table 3).” The revised text can be found in lines 155-156.

Comments 21: On line 139, "titratable acidity" - Do you mean: "The titratable acidity of wine grapes from the N3Fe1 treatment was 6.67 to 18.84% lower than the other treatments"?

Response 21: Yes, we have modified this sentence to read: “The titratable acidity of wine grapes from the N3Fe1 treatment was 6.67 to 18.84% lower than the other treatments.” The revised text can be found in lines 156-157.

Comments 22: On lines 141-150, please rewrite (see suggestion above).

Response 22: Thank you for your suggestion. We have rewritten this section to make it clearer. The revised content can be found in lines 157-164.

Comments 23: Change “application” to “co-application” on line 151.

Response 23: I appreciate your input. The term on line 151 has been updated to “co-application” as suggested.

Comments 24: In Table 3, change “Soluble solids” to “Soluble solids content” or “SSC.”

Response 24: Thank you for your suggestion. We have made the change to "Soluble solids content" in Table 3.

Comments 25: Change “Titrate acid” to “Titratable acidity” in Table 3.

Response 25: Thank you for pointing this out. We have changed the reference to “Titrate acid” in Table 3.

Comments 26: Change “Effect of N and Fe co-application at different stages on relative content of essential amino acids in wine grapes” to “Effect of co-application of N and Fe on wine grape relative content of essential amino acids at different developmental stages.”

Response 26: Thank you for your suggestion. I have updated the text on line 156 to “Effect of co-application of N and Fe on wine grape relative content of essential amino acids at different developmental stages.”

Comments 27: Change “significantly increased the” to “showed a significantly higher content” on line 160.

Response 27: I appreciate your input. The text on line 160 has been revised to “showed a significantly higher content” as recommended.

Comments 28: Change “Effect of N and Fe co-application at different stages on flavonoids in wine grape pericarp” to “Effect of co-application of N and Fe on wine grape yield flavonoids content at different developmental stages.”

Response 28: Thank you for your recommendation. I have revised the text on line 176 to “Effect of co-application of N and Fe on wine grape yield flavonoids content at different developmental stages” as suggested.

Comments 29: On line 178, what do you mean by "flavonoid species" and "flavonoid substances"? Compounds?

Response 29: Thank you for pointing this out. The terms "flavonoid species" and "flavonoid substances" refer to different compounds within the flavonoid family. We acknowledge that the initial presentation was unclear and have standardized the nomenclature.

Flavonoid species: This term refers to the specific types or varieties of flavonoid compounds detected in the wine grape peel. For instance, the analysis identified 33 different flavonoid compounds out of a total of 46 that were screened. Examples of these flavonoid species mentioned in the text include genistein, apigenin, baicalin, and hesperetin.

Flavonoid substances: This term is essentially synonymous with "flavonoid species" in this context and refers to the individual flavonoid compounds that were detected and quantified in the grape.

Comments 30: Change “N or Fe” to “co-applications of N and Fe” on line 206.

Response 30: I appreciate your input. The text on line 206 has been updated to “co-applications of N and Fe” as requested.

Comments 31: Change “Soluble solids and titratable acids” to “Sugars and organic acids” on line 233.

Response 31: Thank you for pointing that out. I have corrected “Soluble solids and titratable acids” to “Sugars and organic acids” on line 233.

Comments 32: Change “most organic acids in grapes” to “many other compounds” on line 234.

Response 32: I appreciate your feedback. The text on line 234 has been revised to “many other compounds” as suggested.

Comments 33: On lines 251-257, you should include references to previously published studies to support your assumptions and data.

Response 33: Thank you for your suggestion. We have supplemented references to previously published studies to support our assumptions and data in lines 251-257.

Comments 34: On lines 299-300, this needs a reference. Where did you obtain these values/information?

Response 34: Thank you for pointing this out. We have supplemented the necessary reference to support the values and information mentioned in lines 299-300.

Comments 35: On line 303, should “training” be “training or draining”?

Response 35: Thank you for your query. In the context of viticulture and grapevine management, the term "training system" is correct. It refers to the methods used to shape and support the grapevines for optimal growth, sunlight exposure, and ease of management. Training System: This involves various techniques and structures used to guide the growth of grapevines, ensuring that the canopy is well-managed and the fruit is exposed to adequate sunlight. Common training systems include the "VSP" (Vertical Shoot Positioning), "Guyot," "Pergola," and in this case, the "sloping frame" system. These systems help improve grape quality and facilitate vineyard management tasks like pruning, spraying, and harvesting.

Comments 36: On line 325, for “electric sprayer,” please provide specifications, name, manufacturer, etc.

Response 36: Thank you for your suggestion. The electric sprayer used is a 20L, backpack-type, electric sprayer with a stirring function, Zhunongli.

Comments 37: On line 333, for "selected leaves," please specify how many.

Response 37: Thank you for your suggestion. We have clarified the exact number of leaves and revised the sentence to: “From each treatment group, ten randomly selected leaves of uniform length and height on the same side of each plant were quickly placed in a sampling box with dry ice and brought back to the laboratory to determine chlorophyll a and chlorophyll b content.”

Comments 38: On line 338, for "crusher," please specify the type of crusher.

Response 38: I have specified the type of crusher used in the manuscript. It is a 1500g, RS-FS1612 crusher, Rongshida.

Comments 39: On line 339, for "bag ged," please specify the type of bag.

Response 39: To address your comment, the specific type of bag used in the study is a 100mm x 150mm transparent sealable bag, manufactured by Weihyu.

Comments 40: On line 340, "and the Fe content of the plants was burned" is not clear.

Response 40: Thank you for your feedback.  We have revised the paragraph for clarity.  The updated sentence now reads: “The Fe content of the plant samples was determined using a rigorous analytical procedure.  Initially, the samples were incinerated in a muffle furnace at 550°C to ensure complete combustion of organic matter.  Subsequently, the residual ash was analyzed using an inductively coupled plasma optical emission spectrometer (ICP-OES, Varian, USA) for precise quantification of the Fe content.”  This revised text can be found in lines 385-389.

Comments 41: On line 348, for "electronic balance," please provide specifics of the balance used.

Response 41: I appreciate your input. The electronic balance used in the study is the TD20002A model by LICHEN, with a capacity of 200g and an accuracy of 0.01g.

Comments 42: On line 349, For "Vernier calipers," please provide more specifications (manufacturer, location, etc.).

Response 42: I appreciate your attention to detail. The Vernier calipers used are model BXGYBKC-220779, manufactured by SYNTEK.

Comments 43: On line 352, for "handheld sugar meter," please provide more specifications.

Response 43: Thank you for pointing that out. The handheld sugar meter used is a range of 0-90% BRIX, manufactured by AIREP.

Comments 44: On line 353, for "grains," do you mean berries or seeds?

Response 44: Thank you for the clarification. We have modified "grains" to "berries".

Comments 45: On line 354, for "powdered," how were they powdered?

Response 45: Thank you for your question. The samples were flash-frozen using liquid nitrogen and then ground into powder.

Comments 46: On line 354, for "certain amount," how much precisely?

Response 46: Thank you for pointing this out. We have specified the amount as 5 grams. This detail has been added to section “4.3 Berry quality, yield, and amino acid and flavonoid content” on line 405.

Comments 47: On line 355, for "acidified methanol," please provide more details.

Response 47: Thank you for your suggestion. We have specified that the methanol was acidified with 0.1% hydrochloric acid (v/v). This information has been included in section “4.3 Berry quality, yield, and amino acid and flavonoid content” on line 406.

Comments 48: On line 355, for "centrifuged," please provide the type of centrifuge (model, manufacturer, etc.).

Response 48: I appreciate your attention to detail. The centrifuge used is the MK-20RB high-low speed refrigerated centrifuge, manufactured by Michael.

Comments 49: On line 355, for "several times," how many times?

Response 49: Thank you for your question. We have specified that the centrifugation was performed three times. This detail has been added to section “4.3 Berry quality, yield, and amino acid and flavonoid content” on line 408.

Comments 50: On line 356, for "ultrasonication," please provide more details.

Response 50: Thank you for your suggestion. We have specified that ultrasonication was performed for 15 minutes at a frequency of 40 kHz and a temperature of 25°C. This information has been added to section “4.3 Berry quality, yield, and amino acid and flavonoid content.”

Comments 51: On line 361, for "repeat picking twice," were the 30 berries harvested twice (2 x 30)? At different dates?

Response 51: Thank you for pointing out the lack of clarity. To clarify, a total of 30 berries were picked, with 15 used for the determination of amino acids and the remaining 15 used for the determination of flavonoids. We have rewritten this section to make it clearer. The revised text can be found in lines 419-421.

Comments 52: On line 364, for "soaking," soaking in what?

Response 52: Thank you for your question. We have specified that the berries were soaked in distilled water. This detail has been added to section “4.3 Berry quality, yield, and amino acid and flavonoid content” on lines 423-425.

Comments 53: On line 368, for "column," please provide more specifications.

Response 53: Thank you for your suggestion. We have specified that the column used was 4.6 mm × 150 mm with a particle size of 3 μm. This detail has been added to section “4.3 Berry quality, yield, and amino acid and flavonoid content” on line 428.

Comments 54: On line 370, for "dried," please provide specifications of the vacuum drier.

Response 54:Thank you for your observation. The samples were dried using a 53L LCDZF-6050AB vacuum drier, manufactured by LICHEN.

Comments 55: On line 371, for "MM400, Retsch," please provide specifications.

Response 55: Thank you for your suggestion. We have specified that the equipment used is the MM400 mixer mill from Retsch GmbH, Haan, Germany. This detail has been added to section “4.3 Berry quality, yield, and amino acid and flavonoid content” on line 432.

Comments 56: On line 374, for "centrifuged," please provide centrifuge specifications.

Response 56: Thank you for your suggestion. We have included the specifications for the centrifuge in section “4.3 Berry quality, yield, and amino acid and flavonoid content.”

Comments 57: On line 375, for "microporous membrane," please provide specifications.

Response 57: Thank you for your suggestion. We have specified that the microporous membrane used is a 0.22 µm PTFE membrane. This detail has been added to section “4.3 Berry quality, yield, and amino acid and flavonoid content” on line 437.

Comments 58: On line 376, for "UPLC-MS/MS," please provide specifications (model, manufacturer, location, etc.).

Response 58: Thank you for your suggestion. We have included the specifications for the UPLC-MS/MS, including the model and manufacturer, in section “4.3 Berry quality, yield, and amino acid and flavonoid content” on line 438.

4. Response to Comments on the Quality of English Language

Response: We have revised Quality of English Language for the full manuscript to ensure better compliance with the journal's requirements.

5. Additional clarifications

Response to Reviewer 1 Comments

1. Summary

2. Questions for General Evaluation

Reviewer’s Evaluation

Response and Revisions

Does the introduction provide sufficient background and include all relevant references?

Yes

Is the research design appropriate?

Yes

Are the methods adequately described?

Must be improved

Are the results clearly presented?

Can be improved

Are the conclusions supported by the results?

Yes

3. Point-by-point response to Comments and Suggestions for Authors

Major comments

Comments 1: lines 92-94 Over all, total leaf chlorophyll was highest under N2Fe1 treatment, at 1.66 mg g-1, which increased by 5.73-144.12% compared to the other treatments, with an overall performance as follows: N2Fe1>N2Fe3>N1Fe2>N1Fe1>N3Fe1>N2Fe2>N1Fe3>N3Fe2>N3Fe3’.The differences in most cases are statistically insignificant according to the presented data.

Response 1: We appreciate the reviewer’s insightful comment. Upon careful review, we identified an inaccuracy in our initial presentation. We have revised the relevant section (lines 105-110) to accurately reflect the data. The revised text now reads:

“Overall, compared to the other treatments, the total chlorophyll content of the leaves under the N2Fe1 treatment was 1.66 mg g-1. This value was higher than that of N1Fe3, N2Fe2, N3Fe2, and N3Fe3 treatments by 8.17-52.29%, 24.64%-28.21%, 23.63%-57.24%, and 48.29%-65.56%, respectively.”

This adjustment ensures clarity and precision in reporting the chlorophyll content differences among the treatments. Thank you for highlighting this issue.

Comments 2: Fe leaf/petiole distribution is rather attractive. In general, not absolute values are significant but ratios often give higher information. May be it will be interesting to evaluate leaf/petiole ratios for Fe (or N)?

Response 2: We appreciate the reviewer's valuable suggestion. We have reprocessed the data to evaluate the leaf/petiole ratios for Fe and N content. Additionally, we have included relevant plots and have provided a thorough analysis and discussion in the “3. Discussion” section. This enhancement adds a deeper understanding of the nutrient distribution patterns. Thank you for this insightful recommendation.

Comments 3: no data of sugar content and method of their determination

Response 3: Thank you for your valuable feedback.  In this study, we have used soluble solids content (SSC) as a representative measure for sugar content.  To enhance clarity and consistency, we have standardized the term “sugar content” to “soluble solids content.”  We would like to provide the following explanation for the relationship between sugar content and SSC:

1.     Measurement Techniques: SSC is commonly measured using refractometers or density meters, which evaluate the refractive index or density of the solution.  Due to their high solubility and concentration, sugars have a significant impact on these measurements.

2.     Prevalence of Sugars in Fruits and Vegetables: In many fruits and vegetables, sugars such as glucose, fructose, and sucrose are the predominant components of soluble solids.  These sugars are synthesized through photosynthesis and are primarily responsible for the sweetness of the fruit.

3.     High Solubility of Sugars: Sugars exhibit a high solubility in water, dissolving completely to form a homogeneous solution.  As a result, sugars are the principal contributors to the soluble solids content measured.

We appreciate your understanding and hope this explanation clarifies the rationale behind our methodology.

Comments 4:indicate the units of Titratable Acidity- % per citric of malic acid?

Response 4: Thank you for your pertinent comment. When determining the titratable acidity of wine grape berries using the sodium hydroxide titration method, the primary acid measured is tartaric acid, which is the predominant organic acid in grapes. Additionally, grapes contain significant amounts of malic acid and minor quantities of citric acid. However, during the titration process, tartaric acid typically constitutes the majority of the titratable acidity, making it the principal acid considered.

To provide a more precise representation, we have incorporated detailed information about the titratable acids into the “4. Materials and Methods” section and updated Table 3 accordingly. This clarification should address the unit specification effectively.

Comments 5: did the treatments change the phenolic distribution between peel and pulp?

Response 5: Thank you for your insightful question. In this experiment, the grape berries were rapidly frozen in liquid nitrogen, then crushed whole in the laboratory and weighed as a powder. Due to this methodology, we did not separate the peel and pulp before analysis, and therefore, we do not have specific data on whether the treatments altered the phenolic distribution between these parts. Future studies could be designed to address this aspect more specifically by analyzing the peel and pulp separately.

Comments 6: tannins and anthocyanins belong to polyphenols. why the amount of tannins in Table 3 is higher than the total phenolics?

Response 6: Thank you for your insightful observation. The reason for this may be because the reagents used for the detection of tannins in the Folin-Denis method may react more strongly or more specifically with tannin compounds, resulting in higher apparent concentrations. In contrast, the Folin-Ciocalteu reagent, although used for the detection of a wide range of phenolics, may not react as strongly with tannins, resulting in lower levels of tannins in the total phenolics. The Folin-Denis method measures the blue color that is formed when tannic acid reacts with phosphotungstic molybdic acid. This method is more sensitive or gives a higher reading for tannins. The Folin-Ciocalteu method measures the blue color complex formed when phenols react with Folin-Ciocalteu reagent. This method covers a wide range of phenolic compounds and may dilute the apparent concentration of tannins in the total phenolic content. ​We have corrected the data and reanalyzed them to ensure accurate representation.

Comments 7: indicate the variations in chlorophyll a/b ratio according to different treatments

Response 7: Thank you for your valuable comment. We have reprocessed the data to evaluate the variations in the chlorophyll a/b ratio under different treatments. Additionally, we have included a figure illustrating these variations and provided a detailed analysis and discussion in the “3. Discussion” section. This enhancement offers a clearer understanding of the impact of the treatments on the chlorophyll a/b ratio.

Comments 8: I should recommend to revise Material and Methods section and give more detailed description on the methods of determination of (i) chlorophyll (no data at all), (ii) phenolics including anthocyanines, tannins, (iii) sugar (no data) and indicate the method of available N, P, K, Fe in soil

Response 8: Thank you for your insightful suggestions. We have revised the Materials and Methods section to include detailed descriptions of the methods used for determining chlorophyll, phenolics (including anthocyanins and tannins), and soil nutrients (N, P, K, Fe).  Additionally, we have provided an explanation of the sugar determination method in our response to Comment 3.

Minor comments

Comments 1: Table 1- decipher all abbreviations in the footnote

Response 1: Thank you for your valuable feedback. We have now added footnotes to Table 1 in line 82 of the manuscript to clarify the abbreviations Pn, Gs, Tr, Ci, and WUE. We appreciate your attention to detail and hope this improves the clarity of our table.

Comments 2: Figure 1.‘N and Fe co-application on the chlorophyll of wine grape leaves’ change to : ‘Effect of N and Fe co-application on the chlorophyll of wine grape leaves

Response 2: Agree, thank you for your suggestion. I have made the change as requested.

Comments 3: Fig.2- add (A) and (B) to the title

Response 3: Thank you for your insightful suggestion. I have added (A) and (B) to the title of Figure 2 as requested.

Comments 4: Table 5 decipher ‘***’

Response 4: Thank you for pointing that out. I have deciphered the ‘***’ in Table 5 as requested.

Comments 5: line 233 ‘Soluble solids and titratable acids are the raw materials required for the synthesis of most organic acids in grapes’- but in general soluble solids is a sum of sugar, minerals and acids- and titratable acids refer to organic  acids- TA- is a sum of organic acids but not raw materials required for their synthesis

Response 5: Thank you for pointing out this inaccuracy. The sentence has been corrected to accurately reflect the composition and role of soluble solids and titratable acids. The revised statement can be found in lines 266-267.

Comments 6: line 340 ‘the Fe content of the plants was burned in a muffle furnace at 550 °C and determined using an inductively coupled plasma optical emission spectrometer’- style revise

Response 6: We appreciate your suggestion.  The sentence has been revised for clarity and precision.  The updated sentence can be found in lines 386-387.

Comments 7: Reference list- use journals’ abbreviations. See author’s guidelines

Response 7: Thank you for your comment. We have revised the reference list to use the appropriate journal abbreviations in accordance with the author’s guidelines.

Comments 8: ref 13,19,33, etc =use Italics for plant species

Response 8: Thank you for your comment. We have revised the references to use italics for plant species as required.

Comments 9: ref 32- add the number of the article

Response 9: Thank you for your comment.  We have updated reference 32 to include the number of the article as required.

Comments 10: Tables 4,5- add units of the parameters

Response 10: Thank you for your valuable suggestion. I have added the units of the parameters in Tables 4 and 5 as requested.

4. Response to Comments on the Quality of English Language

Response: We have revised Quality of English Language for the full manuscript to ensure better compliance with the journal's requirements.

5. Additional clarifications

Round 2

Reviewer 1 Report

Comments and Suggestions for Authors

There are only two comments:

1) Table 1 - decipher Pn, Gs, Tr, Ci and WUE in the footnote. Any data in the table should be understood without the text

2) Table 3- just the same: decipher SSC and TAC in the footnote